# Genomic analysis of male puberty timing highlights shared genetic basis with hair colour and lifespan

Ben Hollis[1,107], Felix R. Day [1,107], Alexander S. Busch [1,2,3], Deborah J. Thompson [4], Ana Luiza G. Soares [5,6], Paul R.H.J. Timmers [6,7], Alex Kwong [5,8,9,10], Doug F. Easton [4], Peter K. Joshi [6,7], Nicholas J. Timpson [5], The PRACTICAL Consortium*, 23andMe Research Team*, Ken K. Ong [1,11,107 ✉] & John R.B. Perry[1,107 ✉]

The timing of puberty is highly variable and is associated with long-term health outcomes. To date, understanding of the genetic control of puberty timing is based largely on studies in women. Here, we report a multi-trait genome-wide association study for male puberty timing with an effective sample size of 205,354 men. We find moderately strong genomic correlation in puberty timing between sexes (rg = 0.68) and identify 76 independent signals for male puberty timing. Implicated mechanisms include an unexpected link between puberty timing and natural hair colour, possibly reflecting common effects of pituitary hormones on puberty and pigmentation. Earlier male puberty timing is genetically correlated with several adverse health outcomes and Mendelian randomization analyses show a genetic association between male puberty timing and shorter lifespan. These findings highlight the relationships between puberty timing and health outcomes, and demonstrate the value of genetic studies of puberty timing in both sexes.

[1] MRC Epidemiology Unit, Institute of Metabolic Science, University of Cambridge School of Clinical Medicine, Cambridge Biomedical Campus Box 285, Cambridge CB2 0QQ, UK. [2] Department of Growth and Reproduction, Rigshospitalet, University of Copenhagen, 2100 Copenhagen O, Denmark. [3] International Center for Research and Research Training in Endocrine Disruption of Male Reproduction and Child Health, Rigshospitalet, University of Copenhagen, 2100 Copenhagen O, Denmark. [4] Centre for Cancer Genetic Epidemiology, Department of Public Health and Primary Care, University of Cambridge, Cambridge, UK. [5] MRC Integrative Epidemiology Unit at the University of Bristol, Bristol, UK. [6] Population Health Science, Bristol Medical School, University of Bristol, Bristol, UK. [7] Usher Institute for Population Health Sciences and Informatics, University of Edinburgh, Teviot Place, Edinburgh EH8 9AG, UK. [8] IUMSP, Biopôle, Secteur Vennes-Bâtiment SV-A, Route de la Corniche 10, 1010 Lausanne, Switzerland. [9] School of Geographical Sciences, University of Bristol, Bristol, UK. [10] Centre for Multilevel Modelling, University of Bristol, Bristol, UK. [11] Department of Paediatrics, University of Cambridge School of Clinical Medicine, Cambridge Biomedical Campus Box 181, Cambridge CB2 0QQ, UK. [107]These authors contributed equally: Ben Hollis, Felix R. Day, Ken K. Ong and John R.B. Perry. *Lists of authors and their affiliations appear at the end of the paper. ✉email: ken.ong@mrc-epid.cam.ac.uk; john.perry@mrc-epid.cam.ac.uk

The timing of puberty varies widely in populations and is determined by a broad range of environmental and genetic factors[1]. Understanding the biological mechanisms underlying such variation is an important step towards understanding why earlier puberty timing is consistently associated with higher risks for a range of later life diseases, including several cancers, cardiovascular disease and Type 2 diabetes[2–4].

Most of our understanding of the genetic determinants of puberty timing is based on studies in women, as the age of first menstrual bleeding (age at menarche, AAM) is a well-recalled and widely measured marker of female sexual development. A recent large-scale genome-wide association study (GWAS) for AAM, in ~370,000 women, identified 389 independent signals, accounting for approximately one quarter of the estimated heritability for the trait[5]. In contrast, genetic studies of puberty timing in men are much fewer and smaller in scale, due to lack of data in many studies on male pubertal milestones. We previously reported a GWAS for recalled age at voice breaking in men ($N = 55,871$) from a single study, 23andMe, which identified 14 genetic independent genetic association signals, and many signals with similar effect sizes on voice breaking as for AAM in women[6]. This overlapping genetic architecture between puberty timing in males (in 23andMe) and females was also reported by others[7], and supports the use of recalled age at voice breaking in men as an informative measure of puberty timing for further genetic studies.

Although the overall shared genetic architecture for pubertal timing between sexes is high ($r_g = 0.74$) and many genetic variants show similar effects sizes in both sexes, there are a number of genetic signals that differ between sexes. Most notably, at the *SIM1/MCHR2* locus, the allele that promotes earlier puberty in one sex delays it in the other[6]. Sex-specific expression patterns for several of such highlighted genes was reported in the hypothalamus and pituitary of pre-pubertal mice[8]. Further work to identify the pubertal mechanisms that are divergent between sexes may shed light on differences in later life disease risk associations between sexes.

Here, we greatly extend our previously reported GWAS for age at voice breaking in 23andMe[6] by combining additional data from the UK Biobank study[9]. In this four-fold larger sample, we increase the number of genomic loci that have male-specific effects on puberty timing from 5 to 29, and identify biological pathways that warrant further investigation.

## Results

### Onset of facial hair as a marker of male puberty timing.
Data on relative age of voice breaking and relative age of first facial hair were available in up to 207,126 male participants in the UK Biobank (UKBB) study. For each of these measures, participants were asked if the event occurred relative to their peers: younger than average, about average or older than average. We previously reported that signals for AAM in women show concordant associations with dichotomised voice breaking traits in these UKBB men[5], but to our knowledge no genetic study has previously evaluated timing of facial hair appearance as a marker of pubertal development.

We defined two dichotomous facial hair variables in UKBB men: (1) relatively early onset ($N = 13,226$) vs. average onset ($N = 161,175$); and (2) relatively late onset ($N = 26,066$) vs. average onset. These facial hair traits were in concordance with similarly dichotomised voice breaking traits defined in the same individuals, although more men reported early or late facial hair onset (39,292; 19.6%) than early or late voice breaking (19,579; 10.2%) (Supplementary Table 1). To test the ability of these facial hair traits to detect puberty timing loci, we assessed their associations with previously reported AAM loci. Of the 328

reported autosomal AAM signals for which genotype data were available in UKBB, 266 (81.1%, binomial-$P = 1.2 \times 10^{-31}$) and 276 (84.1%, binomial-$P = 2.1 \times 10^{-38}$) showed directionally concordant individual associations with relatively early and relatively late facial hair, respectively. Furthermore, substantially more AAM signals showed at least nominally significant associations (GWAS $P < 0.05$) with relatively early (102, 31.1%) or relatively late facial hair (152, 46.3%) compared to ~16 expected by chance for each outcome (Supplementary Data 1).

### A multi-trait GWAS for puberty timing in men.
We analysed four GWAS models in UKBB men (imputation v2, ~7.4M SNPs): two models (early and late) for each of relative timing of voice breaking and facial hair. The shared genetic architecture between these four traits was high (genetic correlations ($r_g$) aligned to later timing of voice breaking ranged 0.57–0.91; Supplementary Table 2), and all four showed high genetic correlation with the continuously measured age at voice breaking in 23andMe ($r_g$ 0.61–0.81). These high correlations supported the rationale to combine the GWAS data across all five strata using MTAG[10]. This approach enables genetic data from correlated traits and overlapping samples to be combined in a single analysis, provided three key assumptions are met: (1) homogeneity of variance-covariance matrix for effect sizes for all SNPs across all traits; (2) sampling variation can be ignored and (3) sample overlap is adequately captured. Simulations involving large numbers of traits with extreme sample overlap have demonstrated that assumptions (2) and (3) may safely be made for most applications of MTAG. Violation of the assumption of homogeneity of effect sizes may be plausible in this setting if some SNPs have an effect on voice breaking and not facial hair (and vice-versa). We therefore calculated the upper bound for the false discovery rate (FDR) in our meta-analysis using the 'maxFDR' calculation developed by the MTAG authors. As the value for maxFDR was $7.9 \times 10^{-4}$, this indicates that test statistics are unlikely to be biased due to any violation of the homogeneity assumption.

We therefore aimed to enhance power to identify GWAS signals for continuous age at voice breaking (as recorded in 23andMe) by using MTAG to combine data from the four dichotomous UKBB puberty timing traits all aligned to later timing of voice breaking; this approach yielded an effective GWAS meta-analysis sample size of 205,354 men for continuous pubertal timing. In this combined MTAG dataset, we identified 7,897 variants associated with male puberty timing at the genome-wide statistical significance threshold (GWAS-$P < 5 \times 10^{-8}$), comprising 76 independent signals (Fig. 1; Supplementary Data 2). The most significantly associated variant (rs11156429, GWAS-$P = 3.5 \times 10^{-52}$) was located in/near *LIN28B*, consistent with previously reported studies in men and women. Of these 76 signals, 29 were not in linkage disequilibrium (LD, conservatively defined as $r^2 > 0.05$) with a previously reported signal for puberty timing (either AAM in women or voice breaking in men) (Table 1).

We sought collective confirmation of the 76 independent signals for male puberty timing in up to 2394 boys with longitudinally assessed pubertal sexual characteristics in The Avon Longitudinal Study of Parents and Children (ALSPAC)[11]. In ALSPAC boys, a polygenic risk score for later puberty timing based on 73 of the male puberty timing signals was associated with older ages at voice breaking, peak height velocity and appearance of armpit hair (Linear regression: $P = 4.5 \times 10^{-3}$ to $P = 1.6 \times 10^{-14}$), and was associated with lower likelihood of attaining various milestones of pubertal maturation at specific time-points ($P_{min}$ (pubic hair development at age 14.7 years old): $= 4.6 \times 10^{-10}$, univariate model $r^2 \sim 1.5\%$)(Supplementary Data 3).

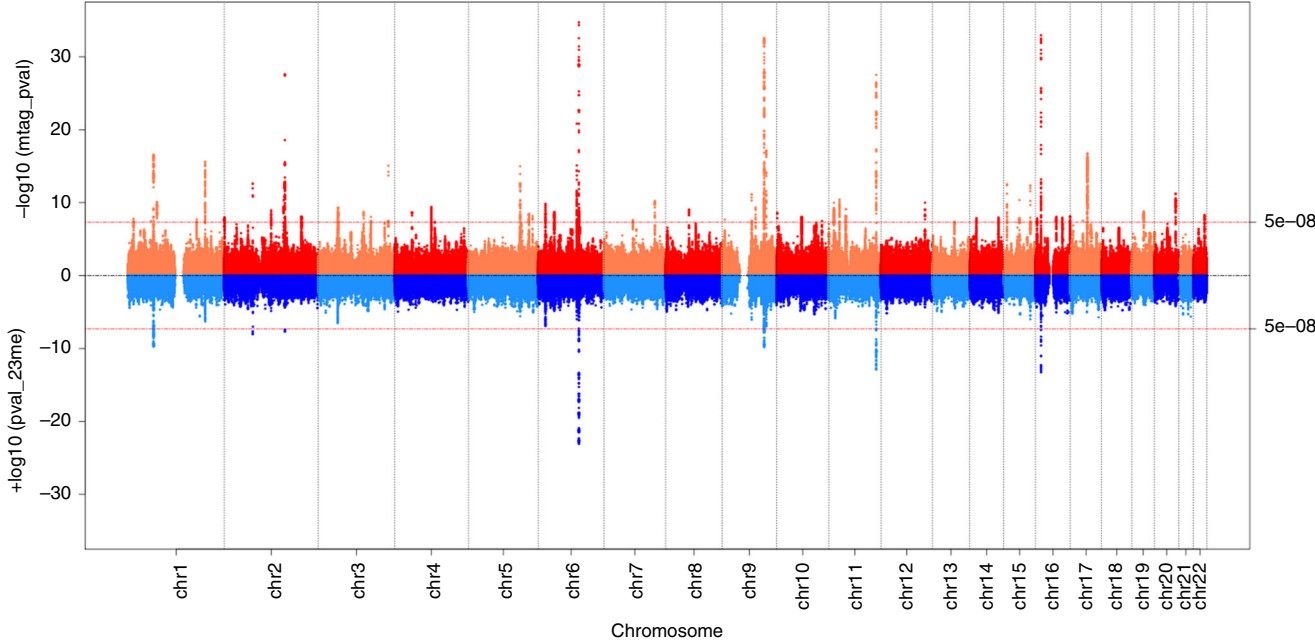

**Fig. 1 Miami plot of meta-analysis for age at voice breaking.** Miami plot comparing $-\log_{10}(P$ values) for SNP associations with age at voice breaking from 23andMe GWAS (bottom half, blue shades) with $-\log_{10}(P$ values) from meta-analysis for age at voice breaking combining 23andMe data with UK Biobank study (top half, red shades). Red line indicates genome-wide significance level ($P < 5 \times 10^{-8}$).

**Table 1 Twenty nine previously unreported signals for puberty timing.**

| Variant | Chr | Position | Alleles | EAF | Nearest gene | Voice breaking Beta (s.e.)[a] | Voice breaking P[a] | Menarche beta (s.e.)[b] | Menarche P[b] |
|---|---|---|---|---|---|---|---|---|---|
| rs71578952 | 7 | 131,001,466 | C/T | 0.495 | MKLN1 | 0.035 (0.003) | $8.4 \times 10^{-28}$ | 0.000 (0.004) | 0.94 |
| rs2222746 | 17 | 44,222,019 | T/G | 0.165 | KIAA1267 | 0.048 (0.004) | $9.0 \times 10^{-28}$ | n/a | n/a |
| rs73182377 | 3 | 181,512,034 | C/T | 0.227 | SOX2OT | 0.040 (0.004) | $1.9 \times 10^{-24}$ | 0.010 (0.005) | 0.05 |
| rs3824915 | 11 | 44,331,509 | G/C | 0.496 | ALX4 | 0.030 (0.003) | $1.0 \times 10^{-20}$ | 0.008 (0.004) | 0.05 |
| rs77578010 | 1 | 11,035,758 | A/G | 0.776 | C1orf127 | 0.036 (0.004) | $1.1 \times 10^{-20}$ | 0.006 (0.005) | 0.18 |
| rs7402990 | 15 | 28,384,491 | G/A | 0.916 | HERC2 | 0.051 (0.006) | $1.4 \times 10^{-18}$ | 0.011 (0.008) | 0.17 |
| rs17833789 | 17 | 55,230,628 | C/A | 0.547 | AKAP1 | 0.028 (0.003) | $5.9 \times 10^{-18}$ | −0.001 (0.004) | 0.83 |
| rs12203592 | 6 | 396,321 | C/T | 0.170 | IRF4 | 0.035 (0.004) | $9.6 \times 10^{-16}$ | 0.002 (0.006) | 0.68 |
| rs35063026 | 16 | 89,736,157 | T/C | 0.069 | C16orf55 | 0.051 (0.006) | $2.1 \times 10^{-15}$ | −0.007 (0.007) | 0.34 |
| rs9690350 | 7 | 547,800 | C/G | 0.419 | PDGFA | 0.025 (0.003) | $3.2 \times 10^{-14}$ | −0.011 (0.008) | 0.16 |
| rs6560353 | 9 | 76,375,544 | G/T | 0.161 | ANXA1 | 0.033 (0.004) | $9.7 \times 10^{-14}$ | 0.016 (0.005) | $2.3 \times 10^{-3}$ |
| rs7905367 | 10 | 54,334,653 | G/C | 0.219 | MBL2 | 0.027 (0.004) | $4.7 \times 10^{-12}$ | −0.011 (0.005) | 0.03 |
| rs7136086 | 12 | 114,129,719 | C/T | 0.734 | RBM19 | 0.025 (0.004) | $1.4 \times 10^{-11}$ | 0.019 (0.005) | $3.2 \times 10^{-5}$ |
| rs10110581 | 8 | 60,691,207 | G/A | 0.263 | CA8 | 0.025 (0.004) | $2.3 \times 10^{-11}$ | 0.014 (0.004) | $1.3 \times 10^{-3}$ |
| rs2842385 | 6 | 19,078,274 | G/A | 0.193 | MIR548A1 | 0.027 (0.004) | $2.5 \times 10^{-11}$ | 0.021 (0.005) | $1.9 \times 10^{-5}$ |
| rs11836880 | 12 | 91,243,529 | G/C | 0.040 | C12orf37 | −0.053 (0.008) | $1.1 \times 10^{-10}$ | 0.006 (0.010) | 0.55 |
| rs10164550 | 2 | 121,159,205 | A/G | 0.657 | INHBB | 0.022 (0.003) | $1.7 \times 10^{-10}$ | 0.016 (0.004) | $1.1 \times 10^{-4}$ |
| rs12895406 | 14 | 36,998,950 | A/G | 0.536 | NKX2-1 | 0.021 (0.003) | $2.8 \times 10^{-10}$ | 0.007 (0.004) | 0.05 |
| rs835648 | 3 | 136,671,504 | A/T | 0.696 | NCK1 | 0.022 (0.004) | $5.2 \times 10^{-10}$ | 0.006 (0.004) | 0.20 |
| rs780094 | 2 | 27,741,237 | T/C | 0.413 | GCKR | 0.020 (0.003) | $7.8 \times 10^{-10}$ | 0.018 (0.004) | $3.7 \times 10^{-6}$ |
| rs1979835 | 5 | 135,689,839 | A/G | 0.876 | TRPC7 | 0.029 (0.005) | $2.5 \times 10^{-9}$ | 0.024 (0.006) | $2.7 \times 10^{-5}$ |
| rs12930815 | 16 | 4,348,635 | C/T | 0.521 | TFAP4 | 0.019 (0.003) | $2.6 \times 10^{-9}$ | 0.008 (0.004) | 0.03 |
| rs12940636 | 17 | 53,400,110 | C/T | 0.655 | HLF | 0.020 (0.003) | $3.6 \times 10^{-9}$ | 0.023 (0.004) | $1.4 \times 10^{-8}$(c) |
| rs60856990 | 17 | 7,337,853 | A/G | 0.629 | TMEM102 | 0.019 (0.003) | $1.3 \times 10^{-8}$ | −0.005 (0.004) | 0.19 |
| rs17193410 | 17 | 32,474,149 | G/A | 0.880 | ACCN1 | 0.028 (0.005) | $1.5 \times 10^{-8}$ | 0.000 (0.007) | 0.98 |
| rs11761054 | 7 | 46,076,649 | C/G | 0.291 | IGFBP3 | −0.020 (0.004) | $2.2 \times 10^{-8}$ | 0.003 (0.004) | 0.51 |
| rs10765711 | 11 | 94,879,318 | C/G | 0.415 | ENDOD1 | 0.018 (0.003) | $2.9 \times 10^{-8}$ | −0.001 (0.004) | 0.81 |
| rs61168554 | 15 | 99,286,980 | A/G | 0.361 | IGF1R | 0.019 (0.003) | $3.8 \times 10^{-8}$ | 0.017 (0.004) | $2.7 \times 10^{-5}$ |
| rs12983109 | 19 | 49,579,710 | G/A | 0.742 | KCNA7 | 0.020 (0.004) | $3.8 \times 10^{-8}$ | 0.002 (0.005) | 0.71 |

Alleles (effect/other), *EAF* effect allele frequency, *Beta* years per allele, *s.e.* standard error, *P* P-value from additive models.
[a]Current data from multi-trait GWAS for age at voice breaking in men.
[b]Previously reported GWAS for age at menarche in women[5].
[c]This signal was not previously highlighted[5] because that study used a distance-based metric to define independent signals rather than a LD based metric.

**Genetic heterogeneity between sexes.** Consistent with our previous study[6], we observed a moderately strong genome-wide genetic correlation in pubertal timing between males and females ($r_g = 0.68$, $P = 2.6 \times 10^{-213}$; based on continuous data on voice breaking and AAM in 23andMe), with similar effect estimates in both sexes for many individual variants (Fig. 2). However, there were exceptions to this overall trend: 5/76 male puberty timing signals (Fig. 2b) and 15/387 reported AAM signals (Fig. 2a) showed significant (by Bonferroni-corrected $P$ values) heterogeneity between sexes in their effects on puberty timing (two of these heterogeneous signals were found in both analyses) (Supplementary Data 4 and 5; Fig. 2). Only one signal showed significant directionally opposite effects (i.e. the allele that conferred earlier puberty timing in one sex delayed puberty in the other sex); rs6931884 at *SIM1/PRDM13/MCHR2*, as previously reported (males: $\beta_{voice-breaking} = -0.064$ years/allele; females:

$\beta_{menarche} = 0.059$ years/allele; $P_{heterogeneity} = 2.6 \times 10^{-14}$). Two variants located near to genes that are disrupted in rare disorders of puberty[12,13] showed no effect or weaker effect in males than in females: rs184950120, 5′UTR to *MKRN3* ($\beta_{voice-breaking} = 0.085$ years/allele; $\beta_{menarche} = 0.396$ years/allele, $P_{heterogeneity} = 3.6 \times 10^{-3}$), and rs62342064, one of 3 AAM variants in/near *TACR3* ($\beta_{voice-breaking} = -0.017$ years/allele; $\beta_{menarche} = 0.057$ years/allele, $P_{heterogeneity} = 4.2 \times 10^{-5}$).

**Implicated genes, tissues and biological pathways.** Two of the 76 lead variants associated with male pubertal timing were non-synonymous: a previously reported AAM signal in *KDM4C* (rs913588), encoding a lysine-specific demethylase, and a male-specific signal in *ALX4* (rs3824915), encoding a homeobox gene involved in fibroblast growth factor (FGF) signalling that is mutated in rare disorders of cranium/central neural system (CNS) development with male-specific hypogonadism[14]. A further 10 lead variants were in strong LD ($r^2 > 0.8$) with one or more non-synonymous variants, of which three have not previously been associated with puberty timing: *FGF11*, which encodes a FGF expressed in the developing CNS and promotes peripheral androgen receptor expression[15], *TFAP4*, which encodes a transcription factor of the basic helix-loop-helix-zipper family[16], and *GCKR*, which encodes a regulatory protein that inhibits glucokinase in liver and pancreatic islets and is associated with a range of cardiometabolic traits[17] (Supplementary Table 3). A further seven are reported signals for AAM, but were not previously reported for voice-breaking. These missense variants are in the following genes: *SRD5A2*, encoding for Steroid 5-alpha-reductase, which converts testosterone into the more potent androgen dihydrotestosterone; *LEPR*, encoding the receptor for appetite and reproduction hormone leptin; *SMARCAD1*, encoding a mediator of histone H3/H4 deacetylation; *BDNF, FNDC9, FAM118A* and *ZNF446*.

Consistent with genetic analyses of AAM in females, tissues in the CNS were the most strongly enriched for genes co-located near to male puberty timing associated variants (Supplementary Figs. 1 and 2; Supplementary Data 6 and Supplementary Table 4). To identify mechanisms that regulate pubertal timing in males, we tested all SNPs genome-wide for enrichment of voice breaking associations with pre-defined biological pathway genes. Four pathways showed evidence of enrichment: histone methyltransferase complex (FDR = 0.01); regulation of transcription (FDR = 0.02)); ATP binding (FDR = 0.03); and cAMP biosynthetic process (FDR = 0.03) (Supplementary Data 7).

**Links between hair colour and puberty timing.** Noting that three loci for puberty timing were located proximal to genes previously associated with pigmentation (*HERC2, IRF4, C16orf55*), we assessed the broader relationship between these traits. It is known that men have darker natural hair colour than women in European ancestry populations[18] and this sex difference appears following the progressive darkening of hair and skin colour during adolescence[19,20]. However, a link between inter-individual variation in natural hair colour and puberty timing has not previously been described. We assessed this phenotypic relationship in up to 179,594 white males of European ancestry in UKBB in a model including 40 genetic principal components (to adjust for even minor subpopulation ethnic variations). Men with red, dark brown and black natural hair colours showed progressively higher odds of early puberty timing, relative to men with blond hair. Similarly, women with darker natural hair colours had earlier puberty timing relative to women with blond hair (Table 2).

To test the shared biological basis between these two phenotypes, we systematically assessed the effects of genetic variants associated with natural hair colour on puberty timing. Using a two-sample

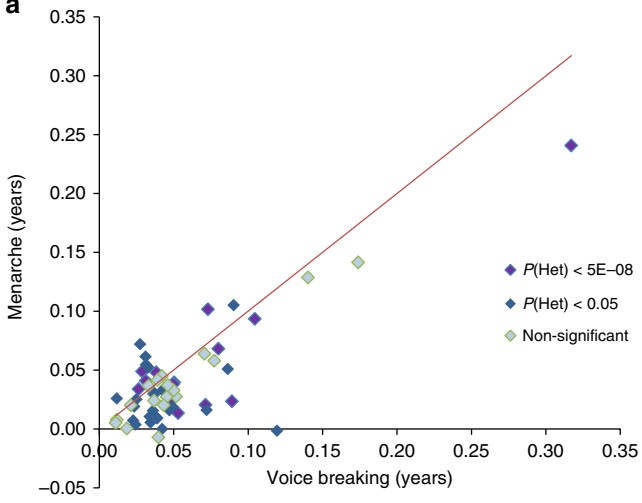

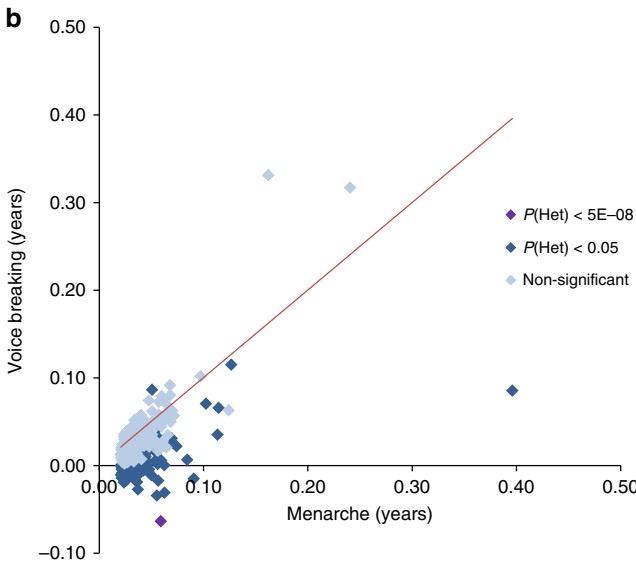

**Fig. 2 Scatterplots comparing variants for effect size on age at menarche and age at voice breaking. a** Scatterplot for independent signals identified in voice breaking meta-analysis (23andMe and UK Biobank) in MTAG. X-axis depicts SNP effect on age at voice breaking in years from 23andMeGWAS; y-axis shows effect on AAM. **b** Scatterplot for known AAM loci depicting effect size for menarche (*x*-axis) and age at voice breaking (*y*-axis). Signals are colour-coded based on heterogeneity P values.

**Table 2 Phenotypic associations between natural hair colour and puberty timing in UK Biobank men and women.**

| Natural hair colour | Effect[a] in men | 95% CI in men | P-value in men | Effect[a] in women | 95% CI in women | P-value in women |
|---|---|---|---|---|---|---|
| Blond | (ref) | | | (ref) | | |
| Red | 1.17 | 1.02, 1.35 | 0.02 | −0.100 | −0.134, −0.066 | $6.2 \times 10^{-9}$ |
| Light brown | 1.00 | 0.91, 1.09 | 0.92 | −0.026 | −0.047, −0.005 | 0.01 |
| Dark brown | 1.45 | 1.34, 1.58 | $6.7 \times 10^{-18}$ | −0.093 | −0.114, −0.072 | $6.7 \times 10^{-18}$ |
| Black | 1.63 | 1.46, 1.81 | $1.2 \times 10^{-19}$ | −0.059 | −0.113, −0.005 | 0.03 |

CI confidence interval.
[a]Effect size in men ($n = 179{,}549$) is odds ratio for early relatively voice breaking (relative to blond hair). Effect size in women ($n = 238{,}195$) is mean difference in age at menarche in years (relative to blond hair).

Mendelian randomisation (MR) approach, we modelled 119 recently reported GWAS hair colour signals as a single instrumental variable[18] on age at voice breaking (in 23andMe men). Such an approach can provide more robust and explicit evidence that there is shared biology between these two traits. The findings infer that susceptibility to darker natural hair colour also confers earlier puberty timing in men ($\beta_{IVW} = -0.044$ years per ordered category, $P_{IVW} = 7.10 \times 10^{-3}$) with no evidence of heterogeneity across signals (Cochrane Q: $P = 0.99$) or directional pleiotropy (MR-Egger Intercept: $P = 0.99$) (Supplementary Fig. 3). Using a similar approach in published AAM data in females[5], we inferred a directionally consistent but weaker association between darker natural hair colour on earlier AAM ($\beta_{IVW} = -0.017$, $P_{IVW} = 3.64 \times 10^{-3}$), and with modest heterogeneity across signals (Cochrane Q: $P = 0.038$) (Supplementary Table 5).

Due to the partial overlapping samples between the SNP instrument discovery and outcome in the above approach, we performed a second, more conservative analysis, using a more limited 5-SNP score aligned to darker hair colour in non-overlapping data on white UK Biobank individuals, adjusting for 40 genetic principal components and geographical location of testing centres. This found highly consistent results in men: the 5-SNP score was associated with higher risk for early voice breaking ($P_{IVW} = 1.72 \times 10^{-19}$) and lower risk for late voice breaking ($P_{IVW} = 5.90 \times 10^{-6}$); but we found no association with AAM in women ($P_{IVW} = 0.23$) (Supplementary Table 5).

**Male puberty timing and other complex traits and diseases**. To assess the extent of shared heritability between male puberty timing and other complex traits, we calculated genome-wide genetic correlations across 751 complex traits/datasets using LD score regression[21] (Supplementary Data 8). Apart from other puberty and growth-related traits, the strongest positive genetic correlation was observed with 'overall health rating', followed by several social traits, including educational attainment, fluid intelligence score and ages at first/last birth. Conversely, male puberty timing showed negative genetic correlations with cardiometabolic diseases, including Type 2 diabetes and hypertension, as well as health risk behaviours, including alcohol intake frequency and smoking (Supplementary Fig. 4). In general, early genetically predicted puberty timing in men was correlated with adverse health outcomes. We then performed MR analyses to test the genetic association with two exemplar traits which have previously been associated with puberty timing: lifespan[22] and prostate cancer[5].

We previously reported genetic associations between earlier AAM and higher risks for hormone sensitive cancers, when adjusting for the protective effects of higher BMI[5,23]. Here, using a similar two-sample multi-variate MR approach, we did not find a significant association with prostate cancer (OR per year = 0.89, 95%CI = 0.79–1.01; $P_{IVW} = 0.06$) or advanced prostate cancer (OR per year = 0.82, 95%CI = 0.67–1.00; $P_{IVW} = 0.05$) (Supplementary

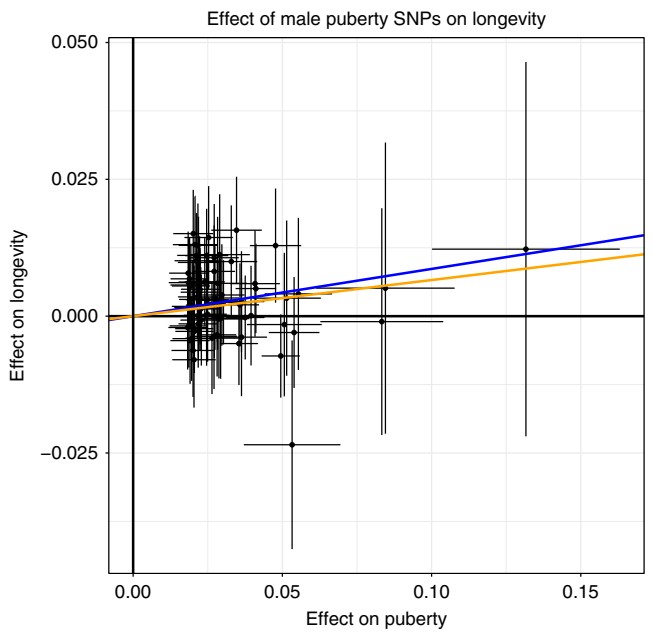

**Fig. 3 Effect of male puberty SNPs on longevity.** Scatterplot of 73 male puberty loci (pruned for heterogeneity) comparing effect sizes for puberty timing (from MTAG) and longevity. Lines show results of different Mendelian randomisation models: IVW (blue), weighted median (yellow). Error bars indicate 95% confidence intervals for individual SNP associations.

Table 6; Supplementary Fig. 6). We used a similar MR approach to test the relevance of male puberty timing genetics to lifespan (see Methods). The findings support a genetic association between later puberty timing in males and longer lifespan, corresponding to 9 months longer life per year later puberty (IVW $P = 6.7 \times 10^{-4}$) (Fig. 3). The result was consistent after removal of one outlier SNP (MR-PRESSO $P = 8.3 \times 10^{-4}$), and with methods robust to the effects of heterogeneous SNPs (weighted median $P = 0.02$) (Supplementary Table 7).

**Discussion**

Here, we report the a genetic study of male puberty timing by performing a meta-analysis across closely related phenotypic traits in two large studies. The effective sample size of 205,354 men is substantially higher than the previous GWAS in men ($n = 55{,}871$), although still smaller than similar studies in women ($n \sim 370{,}000$). While there was overall moderately strong overlap between sexes in the genetic architecture of puberty timing ($r_g = 0.68$), several findings point to mechanisms with particular relevance to men. The genes implicated in puberty timing include, *ALX4* and *SRD5A2*, the disruption of which leads to male

pseudohermaphroditism whereas higher 5-alpha-reductase activity has been reported in women with polycystic ovary syndrome[24], and *INHBB* encoding the beta-B subunit of the hormone Inhibin B, which is secreted by testicular Sertoli cells in males. By contrast, the known female puberty timing locus at *INHBA*, which encodes the beta-A subunit of the hormone Inhibin A, showed little association with puberty timing in males.

Non-genetic observational studies have consistently reported darkening of hair and skin pigmentation in children of European ancestry during the peri-pubertal years[19,20]. Furthermore, the established sexual dimorphism in pigmentation[18] reportedly appears from puberty onwards[19] and the relatively darker skin and hair of men compared to women is postulated to reflect the stronger stimulation of melanogenesis by androgens compared to estrogens[25]. Our findings suggest a more widespread overlap between pigmentation and reproduction, possibly reflecting common regulators of pituitary production of melanocortins and gonadotrophins, or even an impact of melanocyte signalling on puberty timing. The pituitary pro-peptide, pro-opiomelanocortin (POMC), is cleaved into several peptides with significant melanogenic activity (ACTH, α-MSH, and β-MSH) by the pro-hormone convertase enzymes, PC-1 and PC-2, both of which are implicated by previously reported loci for AAM in females[5]. The relationship in women is more complicated suggesting that while there is a relationship between pigmentation and puberty timing in both sexes, it may act in a sex-specific manner.

Finally, our findings substantiate links between puberty timing in men and later life health outcomes, including Type 2 diabetes and hypertension. Our observed genetic association between earlier puberty timing in males and reduced lifespan was consistent across three different MR analytical approaches, which reduces the likelihood of horizontal pleiotropy (where variants influence lifespan though mechanisms separate to puberty timing). Plausible interpretations are a causal effect of earlier male puberty timing on earlier mortality or widespread horizontal pleiotropy (affecting >50% of variants for puberty timing) whereby puberty variants affect biologic pathways, which on the one hand determine puberty timing and on the other hand influence the risk of mortality. We note that horizontal pleiotropy may have been underestimated in this analysis as the same UK Biobank population were included in both exposure and outcome samples[26]. In conclusion, our findings demonstrate the utility of multi-trait GWAS to combine data across studies with related measures to provide insights into the regulation and consequences of puberty timing.

## Methods

**23andMe study**. Genome-wide SNP data were available in up to 55,871 men aged 18 or older of European ancestry from the 23andMe study. Age at voice breaking was determined by response to the question 'How old were you when your voice began to crack/deepen?' in an online questionnaire. Participants chose from one of seven pre-defined age bins (under 9, 9–10 years old, 11–12 years old, 13–14 years old, 15–16 years old, 17–18 years old, 19 years old or older). These were then re-scaled to one year age bins post-analysis by a previously validated method[27]. SNPs were excluded prior to imputation based on the following criteria: Hardy-Weinberg equilibrium $P < 10^{-20}$; call rate < 95%; or a large frequency discrepancy when compared with European 1000 Genomes reference data[6]. Imputation of genotypes was performed using the March 2012 'v3' release of 1000 Genomes reference haplotype panel. Genetic associations with puberty timing were obtained by linear regression models using age and five genetically determined principal components to account for population structure as covariates, with additive allelic effects assumed. P values for SNP associations were computed using likelihood ratio tests. Participants provided informed consent to take part in this research under a protocol approved by Ethical and Independent Review Services, an institutional review board accredited by the Association for the Accreditation of Human Research Protection Programs.

**UK Biobank**. Genotyping for UK Biobank participants has been described in detail previously[9]. For this analysis, we limited our sample to individuals of white

European ancestry. Age at voice breaking for male participants from the UK Biobank cohort study was obtained by responses to the touchscreen question 'When did your voice break?' Participants were required to choose from one of five possible options (younger than average, about average, older than average, do not know, prefer not to answer). For age at first facial hair, respondents were asked to choose from one of the same five options in response to the touchscreen question 'When did you start to grow facial hair?' In total, data were available in up to 191,270 individuals for voice breaking and 198,731 individuals for facial hair. UK Biobank received ethical approval from the NHS National Research Ethics Service North West (11/NW/0382).

Respondents who answered either 'older than average' or 'younger than average' were compared separately using the 'about average' group as a reference in a case-control design for both phenotypes. For both voice breaking and facial hair, relatively early and relatively late effect estimates were obtained using linear mixed models which were applied using BOLT-LMM software, which accounts for cryptic population structure and relatedness. Covariates included age, genotyping chip and 10 principle components.

**Meta-analysis of voice breaking results**. GWAS summary results for each of the five strata (23andMe age at voice breaking; UK Biobank relatively early and late voice breaking; and UK Biobank relatively early and late facial hair) all aligned to later timing of voice breaking were meta-analysed using MTAG[10]. MTAG uses GWAS summary statistics from multiple correlated traits to effectively increase sample size and statistical power to detect genetic associations. Full details on the methodology have been described previously[10]. In brief, MTAG estimates a variance-covariance matrix to correlate the effect sizes of each trait using a moments-based method, with each trait and genotype standardised to have a mean of zero and variance of one. In addition, MTAG calculates a variance-covariance matrix for the GWAS estimation error using LD score regressions. The effect estimate for the association of each SNP on the trait of interest is then derived using a moments-based function, in a generalisation of standard inverse-variance weighted meta-analysis.

Prior to meta-analysis, we removed extremely rare variants (MAF < 0.01). In addition, for the four UK Biobank strata we calculated the effective sample size using

$$N_{eff} = \frac{2}{\frac{1}{N_{cases}} + \frac{1}{N_{controls}}}. \qquad (1)$$

Effective sample sizes for early and late voice breaking were 15,711 and 21,217, respectively, and 17,391 and 23,011 for early and late facial hair, respectively. We used the genome-significant P-value threshold of $P < 5 \times 10^{-8}$ to determine significant SNP associations. Independent signals were identified using distance-based clumping, with the SNP with the lowest P-value within a 1 MB window being considered the association signal at that locus.

**Gene annotation and identification of loci**. For each independent signal identified in the voice breaking meta-analysis, we identified all previously reported age at menarche (AAM) and voice breaking (VB) loci within 1MB of that SNP. We assessed if each locus had previously been associated with puberty timing (AAM or VB) within 1MB, and if any such loci within 1MB were in LD ($r^2 < 0.05$). Heterogeneity between AAM and VB for each SNP was determined by the $I^2$ statistic and P-value generated by the METAL software.

Gene annotation was performed using a combination of methods. Information on the nearest annotated gene was obtained from HaploReg v4.1. In addition, other genes in the region were identified using plots produced from LocusZoom. The most likely causal variant was determined by combining this information with identification of any non-synonymous variants within the region as well as application of existing knowledge.

**Replication in ALSPAC**. The Avon Longitudinal Study of Parents and Children (ALSPAC) recruited pregnant women resident in the Avon area of the UK with an expected delivery date between 1 April 1991 and 31 December 1992. Since then, mothers, partners, and offspring have been followed up regularly through questionnaires and clinical assessments[11]. The offspring cohort consists of 14,775 live-born children (75.7% of the eligible live births). Full details of recruitment, follow-up and data collection have been reported previously[11]. Ethical approval for the study was obtained from the ALSPAC Ethics and Law committee and the Local Research Ethics Committees. A series of nine postal questionnaires regarding pubertal development was administered approximately annually from the time the participant was aged 8 until he was aged 17. The questionnaires, which were responded by either the parents or the participant, had schematic drawings and verbal descriptions of secondary sexual characteristics (genitalia and pubic hair development) based on the Tanner staging system, as well as information on armpit hair growth and voice change. Age at voice change was considered the age at which the adolescent reported his voice to be occasionally a lot lower or to have changed completely. Weight and height were measured annually up to age 13 years, then at ages 15 and 17 years by a trained research team. Age at peak height velocity (PHV) was estimated using Superimposition by Translation And Rotation (SITAR) mixed effects growth curve analysis[28]. The sample size available varied according to

the phenotype, from 1126 (for genital development) to 2403 (for age at which armpit hair started to grow).

The genetic risk score (GRS) was calculated based on 73 SNPs (genotypes at 3 SNPs were unavailable) weighted by the effect size reported for that SNP in the ReproGen Consortium. The GRS was standardised, and results are presented as increase in the phenotype per standard-deviation increase in the GRS. Linear (continuous phenotype) and logistic (binary phenotype) regression analyses were performed unadjusted and adjusted for age (except for age at PHV, age at voice change and age at which armpit hair started to grow) and controlled for population stratification using the first 10 principal components.

**Gene expression and pathway analysis**. We used MAGENTA to investigate whether genetic associations in the meta-analysed dataset showed enrichment in any known biological pathways. MAGENTA has previously been described in detail[29]. In brief, genes are mapped to an index SNP based on a 150 kb window, with a regression model applied to correct the $P$-value (gene score) for gene size, SNP density and LD-related properties. Gene scores are ranked, and the numbers of gene scores observed in a given pathway in the 75th and 95th percentiles are calculated. A $P$-value for gene-set enrichment analysis (GSEA) is calculated by comparing these values to one million randomly generated gene sets. Testing was completed on 3216 pathways from four databases (PANTHER, KEGG, Gene Ontology and Ingenuity). Significance was determined based on an FDR < 0.05 for genes in the 75th or 95th percentile.

To determine tissue-specific expression of genes, we used information from the GTEx project. GTEx characterises transcription levels of RNA in a variety of tissue and cell types, using sample from over 1000 deceased individuals of European, African-American and Asian descent. We investigated transcription levels of significant genes identified in our meta-analysis of voice breaking in 53 different tissue types. We used a conservative Bonferroni-corrected $P$-value of $9.4 \times 10^{-4}$ (=0.05/53) to determine significance.

**Association between hair colour and puberty timing**. Information on natural hair colour for UK Biobank participants was collected via touchscreen questionnaire, in response to the question 'What best describes your natural hair colour? (If your hair colour is grey, the colour before you went grey)'. Participants chose from one of 6 possible colours: blond, red, light brown, dark brown, black or other. For our analyses, we restricted this to include only non-related individuals of white European ancestry, totalling 190,845 men and 238,179 women. Hair colours were assigned numerical values from lightest (blond) to darkest (black) in order to perform ordered logistic regression of hair colour for both relative age at voice breaking in men and AAM in women. In both cases, blond hair was used as the reference group and models were adjusted for the top 10 principle components to account for population structure. In men this produces an effect estimate as an odds ratio for early puberty (relative to blond-haired individuals), while in women the effect estimate is on a continuous scale for AAM (in years) relative to the mean AAM for those with blond hair.

**Genetic correlations**. Genetic correlations ($r_g$) were calculated between age of puberty in males and 751 health-related traits which were publically available from the LD Hub database using LD Score Regression[21,30].

**Mendelian randomisation analyses**. GWAS summary statistics for longevity were obtained from Timmers et al.[31]. Briefly, Timmers et al. performed a GWAS of parent survivorship under the Cox proportional hazard model in 1,012,050 parent lifespans of unrelated subjects using methods of Joshi et al.[32], but extending UK Biobank data to that of second release. Data were meta-analysed using inverse-variance meta-analysis results from UK Biobank genomically British, Lifegen, UK Biobank self-reported British (but not identified as genomically British), UK Biobank Irish, and UK Biobank other white European descent. The resultant hazard ratios and their standard errors were then taken forward for two-sample MR using the 74 male puberty loci which were available in both datasets. We then performed a sensitivity analysis after removing outlier SNPs on the basis of heterogeneity using MR-PRESSO[33].

Summary statistics for the association between the genetic variants and risk of prostate cancer were obtained from the PRACTICAL/ELLIPSE consortium, based on GWAS analyses of 65,044 prostate cancer cases and 48,344 controls (all of European ancestry) genotyped using the iCOGS or OncoArray chips[34]. The analyses were repeated using summary statistics from a comparison of the subset of 9,640 cases with advanced disease versus 45,704 controls, where advanced cases were defined as those with at least one of: Gleason score 8+, prostate cancer death, metastatic disease or PSA > 100. Two-sample MR analyses were conducted by weighted linear regressions of the SNP-prostate cancer log odds ratios (logOR) on the SNP-puberty beta coefficients, using the variance of the logORs as weights. This is equivalent to an inverse-variance weighted meta-analysis of the variant-specific causal estimates. Because of evidence of over-dispersion (i.e. heterogeneity in the variant-specific causal estimates), the residual standard error was estimated, making this equivalent to a random-effects meta-analysis. Unbalanced horizontal pleiotropy was tested based on the significance of the intercept term in MR-Egger

regression. The total effect of puberty timing on prostate cancer risk was separated into a direct effect (independent of BMI, a potential mediator) and an indirect effect (operating via BMI), as described in Burgess et al.[19].

Genetic associations between hair colour and puberty timing were assessed in two ways. First, we performed a two-sample MR analysis based on summary statistics (as described in Burgess et al.[19]) using the most recently reported GWAS for hair colour[18] and 23andMe data for SNP effect estimates on age at voice breaking in men, and published ReproGen consortium data on AAM in women. However, there is partial overlap (between samples used for the discovery phase GWAS of hair colour variants ($n = 290,891$ from 23andMe plus UK Biobank) and the samples used for puberty timing ($n = 55,871$ men from 23andMe; maximum potential overlap = 55,871/290,891 = 19%). Therefore, we also performed a sensitivity analysis in non-overlapping samples; we used a more limited 5-SNP instrument for darker hair colour (identified in an earlier hair colour GWAS that did not include UK Biobank[35]), and assessed its effects on puberty timing in UK Biobank men and women in an individual-level MR analysis, controlling for geographical (assessment centre) and genetic ancestry factors (40 principal components).

**Reporting summary**. Further information on research design is available in the Nature Research Reporting Summary linked to this article.

## Data availability
Puberty associations: The results for the top 10000 SNPs detailed in this manuscript can be found at this https://doi.org/10.17863/CAM.46703. UK Biobank: Data are available to all bona fide researchers for all types of health-related research that is in the public interest, without preferential or exclusive access for any person. All researchers, whether in universities, charities, government agencies or commercial companies, and whether based in the UK or abroad, will be subject to the same application process and approval criteria. 23andMe: Full summary statistics for 23andMe datasets will be made available to qualified researchers under an agreement that protects participant privacy. Researchers should visit https://research.23andme.com/dataset-access/ for more details and instructions for applying for access to the data. ALSPAC: Please note that the study website contains details of all the data that is available through a fully searchable data dictionary and variable search tool at http://www.bristol.ac.uk/alspac/researchers/our-data/.

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

## Acknowledgements

This work was funded by the UK Medical Research Council (MRC; Unit programme MC_UU_12015/2) and used UK Biobank data under application 9905. The MRC, Wellcome Trust (Grant ref: 102215/2/13/2) and the University of Bristol provide core support for ALSPAC. This publication is the work of the authors and <John Perry and Ken Ong> will serve as guarantors for the contents of this paper. GWAS data for ALSPAC was generated by Sample Logistics and Genotyping Facilities at Wellcome Sanger Institute and LabCorp (Laboratory Corporation of America) using support from 23andMe. We are extremely grateful to all the families who took part in the ALSPAC study, the midwives for their help in recruiting them, and the whole ALSPAC team, which includes interviewers, computer and laboratory technicians, clerical workers, research assistants, volunteers, managers, receptionists and nurses. We thank the research participants and employees of 23andMe for making this work possible. Genotyping of the OncoArray was funded by the US National Institutes of Health (NIH) [U19 CA154 148537 for ELucidating Loci Involved in Prostate cancer SuscEptibility (ELLIPSE) project and X01HG007492 to the Center for Inherited Disease Research (CIDR) under contract number 156 HHSN268201200008I]. Additional analytic support was provided by NIH NCI U01 CA188392 157 (PI: Schumacher). THE PRACTICAL consortium was supported by Cancer Research UK Grants C5047/A7357, C1287/A10118, C1287/A16563, C5047/A3354, C5047/A10692, C16913/A6135, European 160 Commission's Seventh Framework Programme grant agreement n° 223175 (HEALTH-F2-2009-223175), and The National Institute of Health (NIH) Cancer Post-Cancer GWAS initiative grant: No. 1 U19 CA 148537-01 (the GAME-ON initiative). We would also like to thank the following for funding support: The Institute of Cancer Research and The Everyman Campaign, The Prostate Cancer Research Foundation, Prostate Research Campaign UK (now Prostate Action), The Orchid Cancer Appeal, The National Cancer Research Network UK, The National Cancer Research Institute (NCRI) UK. We are grateful for support of NIHR funding to the NIHR Biomedical Research Centre at The Institute of Cancer Research and The Royal Marsden NHS Foundation Trust. In memoriam: Brian E. Henderson.

## Author contributions

B.H., F.R.D., K.K.O. and J.R.B.P. designed this study. D.F.E., The PRACTICAL Consortium, the 23andMe Research Team, N.J.T., K.K.O., and J.R.B.P. acquired the data. B.H., F.R.D., A.S.B., D.J.T., A.L.G.S., P.R.H.J.T., A.K., P.K.J, the 23andMe Research Team and J.R.B.P. analysed the data. B.H., F.R.D., A.S.B., K.K.O. and J.R.B.P. interpreted the data and B.H., K.K.O. and J.R.B.P. wrote the first draft of the manuscript. All authors participated in the preparation of the manuscript by reading and commenting on drafts prior to submission.

## Competing interests

The 23andMe Research Team are employees of and own stock or stock options in 23andMe, Inc. The remaining authors declare no conflict of interest.

## Additional information

## The PRACTICAL Consortium

Rosalind A. Eeles[12,13], Brian E. Henderson[14,108], Christopher A. Haiman[14], Zsofia Kote-Jarai[12], Fredrick R. Schumacher[15,16], Ali Amin Al Olama[17,18], Sara Benlloch[12,17], Kenneth Muir[19,20], Sonja I. Berndt[21], David V. Conti[14], Fredrik Wiklund[22], Stephen Chanock[21], Susan Gapstur[23], Victoria L. Stevens[23], Catherine M. Tangen[24], Jyotsna Batra[25,26], Judith Clements[25,26], Australian Prostate Cancer BioResource (APCB), Henrik Gronberg[22], Nora Pashayan[27,28], Johanna Schleutker[29,30], Demetrius Albanes[21], Alicja Wolk[31,32], Catharine West[33], Lorelei Mucci[34], Géraldine Cancel-Tassin[35,36], Stella Koutros[21], Karina Dalsgaard Sorensen[37,38], Eli Marie Grindedal[39], David E. Neal[40,41,42,43], Freddie C. Hamdy[42,43], Jenny L. Donovan[44], Ruth C. Travis[45], Robert J. Hamilton[46], Sue Ann Ingles[14], Barry S. Rosenstein[47,48], Yong-Jie Lu[49], Graham G. Giles[50,51], Adam S. Kibel[52], Ana Vega[53], Manolis Kogevinas[54,55,56,57], Kathryn L. Penney[58], Jong Y. Park[59], Janet L. Stanford[60,61], Cezary Cybulski[62], Børge G. Nordestgaard[63,64], Hermann Brenner[65,66,67], Christiane Maier[68], Jeri Kim[69], Esther M. John[70,71], Manuel R. Teixeira[72,73], Susan L. Neuhausen[74], Kim De Ruyck[75], Azad Razack[76], Lisa F. Newcomb[60,77], Davor Lessel[78], Radka Kaneva[79], Nawaid Usmani[80,81], Frank Claessens[82], Paul A. Townsend[83], Manuela Gago-Dominguez[84,85], Monique J. Roobol[86], Florence Menegaux[87], Kay-Tee Khaw[88], Lisa Cannon-Albright[89,90], Hardev Pandha[91] & Stephen N. Thibodeau[92]

[12]The Institute of Cancer Research, London, UK. [13]Royal Marsden NHS Foundation Trust, London, UK. [14]Department of Preventive Medicine, Keck School of Medicine, University of Southern California/Norris Comprehensive Cancer Center, Los Angeles, CA, USA. [15]Department of Epidemiology and Biostatistics, Case Western Reserve University, Cleveland, OH, USA. [16]Seidman Cancer Center, University Hospitals, Cleveland, OH, USA. [17]Centre for Cancer Genetic Epidemiology, Department of Public Health and Primary Care, University of Cambridge, Strangeways Research Laboratory, Cambridge, UK. [18]University of Cambridge, Department of Clinical Neurosciences, Cambridge, UK. [19]Division of Population Health, Health Services Research and Primary Care, University of Manchester, Manchester, UK. [20]Warwick Medical School, University of Warwick, Coventry, UK. [21]Division of Cancer Epidemiology and Genetics, National Cancer Institute, NIH, Bethesda, MD, USA. [22]Department of Medical Epidemiology and Biostatistics, Karolinska Institute, Stockholm, Sweden. [23]Epidemiology Research Program, American Cancer Society, 250 Williams Street, Atlanta, GA, USA. [24]SWOG Statistical Center, Fred Hutchinson Cancer Research Center, Seattle, WA, USA. [25]Australian Prostate Cancer Research Centre-Qld, Institute of Health and Biomedical Innovation and School of Biomedical Science, Queensland University of Technology, Brisbane, Queensland, Australia. [26]Translational Research Institute, Brisbane, Queensland, Australia. [27]University College London, Department of Applied Health Research, London, UK. [28]Centre for Cancer Genetic Epidemiology, Department of Oncology, University of Cambridge, Strangeways Laboratory, Cambridge, UK. [29]Department of Medical Biochemistry and Genetics, Institute of Biomedicine, University of Turku, Turku, Finland. [30]Tyks Microbiology and Genetics, Department of Medical Genetics, Turku University Hospital, Turku, Finland. [31]Division of Nutritional Epidemiology, Institute of Environmental Medicine, Karolinska Institutet, Sweden. [32]Department of Surgical Sciences, Uppsala University, Uppsala, Sweden. [33]Division of Cancer Sciences, University of Manchester, Manchester Academic Health Science Centre, Radiotherapy Related Research, Manchester NIHR Biomedical Research Centre, TheChristie Hospital NHS Foundation Trust, Manchester, UK. [34]Department of Epidemiology, Harvard T.H Chan School of Public Health, Boston, MA, USA. [35]CeRePP, Tenon Hospital, Paris, France. [36]UPMC Sorbonne Universites, GRC N°5 ONCOTYPE-URO, Tenon Hospital, Paris, France. [37]Department of Molecular Medicine, Aarhus University Hospital, Aarhus, Denmark. [38]Department of Clinical Medicine, Aarhus University, Aarhus, Denmark. [39]Department of Medical Genetics, Oslo University Hospital, Oslo, Norway. [40]University of Cambridge, Department of Oncology, Addenbrooke's Hospital, Cambridge, UK. [41]Cancer Research UK Cambridge Research Institute, Li Ka Shing Centre, Cambridge, UK. [42]Nuffield Department of Surgical Sciences, University of Oxford, Oxford, USA. [43]Faculty of Medical Science, University of Oxford, John Radcliffe Hospital, Oxford, UK. [44]School of Social and Community Medicine, University of Bristol, Bristol, UK. [45]Cancer Epidemiology Unit, Nuffield Department of Population Health University of Oxford, Oxford, UK. [46]Department of Surgical Oncology, Princess Margaret Cancer Centre, Toronto, Canada. [47]Department of Radiation Oncology, Icahn School of Medicine at Mount Sinai, New York, NY, USA. [48]Department of Genetics and Genomic Sciences, Icahn School of Medicine at Mount Sinai, New York, NY, USA. [49]Centre for Molecular Oncology, Barts Cancer Institute, Queen Mary University of London, John Vane Science Centre, London, UK. [50]Cancer Epidemiology & Intelligence Division, The Cancer Council Victoria, Melbourne, Victoria, Australia. [51]Centre for Epidemiology and Biostatistics, Melbourne School of Population and Global Health, The University of Melbourne, Melbourne, Australia. [52]Division of Urologic Surgery, Brigham and Womens Hospital, Boston, MA, USA. [53]Fundación Pública Galega de Medicina Xenómica-SERGAS, Grupo de Medicina Xenómica, CIBERER, IDIS, Santiago de, Compostela, Spain. [54]Centre for Research in Environmental Epidemiology (CREAL), Barcelona Institute for Global Health (ISGlobal), Barcelona, Spain. [55]CIBER Epidemiología y Salud Pública (CIBERESP), Madrid, Spain. [56]IMIM (Hospital del Mar Research Institute), Barcelona, Spain. [57]Universitat Pompeu Fabra (UPF), Barcelona, Spain. [58]Channing Division of Network Medicine, Department of Medicine, Brigham and Women's Hospital/Harvard Medical School, Boston, MA, USA. [59]Department of Cancer Epidemiology, Moffitt Cancer Center, Tampa, USA. [60]Division of Public Health Sciences, Fred Hutchinson Cancer Research Center, Seattle, WA, USA. [61]Department of Epidemiology, School of Public Health, University of Washington, Seattle, WA, USA. [62]International Hereditary Cancer Center, Department of Genetics and Pathology, Pomeranian Medical University, Szczecin, Poland. [63]Faculty of Health and Medical Sciences, University of Copenhagen, Copenhagen, Denmark. [64]Department of

Clinical Biochemistry, Herlev and Gentofte Hospital, Copenhagen University Hospital, Herlev, Denmark. [65]Division of Clinical Epidemiology and Aging Research, German Cancer Research Center (DKFZ), Heidelberg, Germany. [66]German Cancer Consortium (DKTK), German Cancer Research Center (DKFZ), Heidelberg, Germany. [67]Division of Preventive Oncology, German Cancer Research Center (DKFZ) and National Center for Tumor Diseases (NCT), Heidelberg, Germany. [68]Institute for Human Genetics, University Hospital Ulm, Ulm, Germany. [69]The University of Texas MD Anderson Cancer Center, Department of Genitourinary Medical Oncology, Houston, TX, USA. [70]Cancer Prevention Institute of California, Fremont, CA, USA. [71]Department of Health Research & Policy (Epidemiology) and Stanford Cancer Institute, Stanford University School of Medicine, Stanford, CA, USA. [72]Department of Genetics, Portuguese Oncology Institute of Porto, Porto, Portugal. 120 Biomedical Sciences Institute (ICBAS), University of Porto, Porto, Portugal. [73]Biomedical Sciences Institute (ICBAS), University of Porto, Porto, Portugal. [74]Department of Population Sciences, Beckman Research Institute of the City of Hope, Duarte, CA, USA. [75]Ghent University, Faculty of Medicine and Health Sciences, Basic Medical Sciences, Gent, Belgium. [76]Department of Surgery, Faculty of Medicine, University of Malaya, Kuala Lumpur, Malaysia. [77]Department of Urology, University of Washington, Seattle, WA, USA. [78]Institute of Human Genetics, University Medical Center Hamburg-Eppendorf, Hamburg, Germany. [79]Molecular Medicine Center, Department of Medical Chemistry and Biochemistry, Medical University, Sofia, Bulgaria. [80]Department of Oncology, Cross Cancer Institute, University of Alberta, Edmonton, Alberta, Canada. [81]Division of Radiation Oncology, Cross Cancer Institute, Edmonton, Alberta, Canada. [82]Molecular Endocrinology Laboratory, Department of Cellular and Molecular Medicine, KU Leuven, Leuven, Belgium. [83]Institute of Cancer Sciences, Manchester Cancer Research Centre, University of Manchester, Manchester Academic Health Science Centre, St Mary's Hospital, Manchester, UK. [84]Genomic Medicine Group, Galician Foundation of Genomic Medicine, Instituto de Investigacion Sanitaria de Santiago de Compostela (IDIS), Complejo Hospitalario Universitario de Santiago, Servicio Galego de Saúde, SERGAS, Santiago De, Compostela, Spain. [85]University of California San Diego, Moores Cancer Center, La Jolla, CA, USA. [86]Department of Urology, Erasmus University Medical Center, Rotterdam, the Netherlands. [87]Cancer & Environment Group, Center for Research in Epidemiology and Population Health (CESP), INSERM, University Paris-Sud, University Paris-Saclay, Villejuif, France. [88]Clinical Gerontology Unit, University of Cambridge, Cambridge, UK. [89]Division of Genetic Epidemiology, Department of Medicine, University of Utah School of Medicine, Salt Lake City, Utah, USA. [90]George E. Wahlen Department of Veterans Affairs Medical Center, Salt Lake City, UT, USA. [91]The University of Surrey, Guildford, Surrey, UK. [92]Department of Laboratory Medicine and Pathology, Mayo Clinic, Rochester, MN, USA. [108]Deceased: Brian E. Henderson.

## Australian Prostate Cancer BioResource (APCB)

Wayne Tilley[93], Gail P. Risbridger[94,95], Judith Clements[96,97], Lisa Horvath[98,99], Renea Taylor[95,100], Vanessa Hayes[101], Lisa Butler[102], Trina Yeadon[103,104], Allison Eckert[103,104], Pamela Saunders[105], Anne-Maree Haynes[99,101], Melissa Papargiris[94], Srilakshmi Srinivasan[103,104], Mary-Anne Kedda[103,104], Leire Moya[96,97] & Jyotsna Batra[96,97]

[93]Dame Roma Mitchell Cancer Research Centre, University of Adelaide, Adelaide, South Australia, Australia. [94]Monash Biomedicine Discovery Institute Cancer Program, Prostate Cancer Research Program, Department of Anatomy and Developmental Biology, Monash University, Victoria, Australia. [95]Cancer Research Division, Peter MacCallum Cancer Centre, Melbourne, Australia. [96]Institute of Health and Biomedical Innovation and School of Biomedical Sciences, Queensland University of Technology, Brisbane, Queensland, Australia. [97]Australian Prostate Cancer Research Centre-Qld, Translational Research Institute, Brisbane, Queensland, Australia. [98]Chris O'Brien Lifehouse (COBLH), Camperdown, New South Wales, Australia. [99]Garvan Institute of Medical Research, Sydney, New South Wales, Australia. [100]Monash Biomedicine Discovery Institute Cancer Program, Prostate Cancer Research Program, Department of Physiology, Monash University, Clayton, Victoria, Australia. [101]The Kinghorn Cancer Centre (TKCC) Victoria, Victoria, NSW, Australia. [102]Prostate Cancer Research Group, South Australian Health & Medical Research Institute, Adelaide, SA, Australia. [103]Translational Research Institute, Brisbane, Queensland, Australia. [104]Australian Prostate Cancer Research Centre-Qld, Institute of Health and Biomedical Innovation and School of Biomedical Science, Queensland University of Technology, Brisbane, Queensland, Australia. [105]University of Adelaide, North Terrace, Adelaide, South Australia, Australia

## 23andMe Research Team

Michelle Agee[106], Babak Alipanahi[106], Adam Auton[106], Robert K. Bell[106], Katarzyna Bryc[106], Sarah L. Elson[106], Pierre Fontanillas[106], Nicholas A. Furlotte[106], David A. Hinds[106], Karen E. Huber[106], Aaron Kleinman[106], Nadia K. Litterman[106], Matthew H. McIntyre[106], Joanna L. Mountain[106], Elizabeth S. Noblin[106], Carrie A.M. Northover[106], Steven J. Pitts[106], J. Fah Sathirapongsasuti[106], Olga V. Sazonova[106], Janie F. Shelton[106], Suyash Shringarpure[106], Chao Tian[106], Joyce Y. Tung[106], Vladimir Vacic[106] & Catherine H. Wilson[106]

[106]23andMe, Inc., 899 W. Evelyn A 174 venue, Mountain View, CA 94041, USA

