## [Peer Review File · Nature Communications]

Reviewers' Comments:

Reviewer #1:

Remarks to the Author:

Hollis and colleagues describe an interesting GWAS and MR analysis of pubertal timing in males. This is a novel study, using relatively new methods, as most previous studies have focused on pubertal timing in females using age at menarche. They identify 29 new loci for pubertal timing and show some heterogeneity between male and female pubertal timing. Although the manuscript is written well, and is likely to be relevant to a broad range of readers, I have a few issues that are outlined below.

- The questions asked of the UK Biobank participants seem very broad. Were they given any indication of what 'average' was? Did you use any of the data from the follow-ups to provide some sort of validation of their answer? If so, how did you account for the individuals that changed answers?
- The SNP effect estimates from the analysis of early voice breaking and early facial hair in the UK Biobank will be in the opposite direction to the other traits. What impact does this have on the MTAG analysis? Would you get the same result if all of the effect estimates were aligned to later pubertal timing?
- I was under the impression that MTAG reported the adjusted estimates for all traits included in the analysis, or performed an inverse-variance weighted meta-analysis that accounts for sample overlap in the GWAS results assuming that the GWAS are all measuring the same phenotype. What was used for this analysis?
- The MTAG method will be somewhat new to many readers and makes some fairly strong assumptions. Could some discussion of these assumptions of MTAG be included, and whether the authors think they are (or aren't) justified in their analysis?
- Were the FDR calculations conducted that are recommended by the MTAG authors?
- What is the adj_r^2 presented in supplementary table 5 for ALSPAC results? If this is the adjusted r^2 from the full model, then I think it is slightly misleading as it is the variance explained by all covariates in the model, adjusting for the number of covariates, not just by the allele score.
- Would the authors please provide some interpretation of the ALSPAC results, particularly regarding the direction of effect for each phenotype? For example, the beta coefficients for the voice breaking phenotype are negative in the early phases but positive in the latter two and positive for age at voice breaking.
- A number of large GWAS studies with smaller replication samples are using the slope of a regression analysis between the beta coefficients as evidence for replication (see for example the latest GIANT GWAS, PMID: 30124842). Would a similar analysis be useful in ALSPAC, where the age of voice breaking and facial hair appearance measures are more precisely measured, to complement the existing analysis and provide further evidence of whether the SNPs replicate?
- Why were the 23andMe voice breaking data used for the genetic heterogeneity analysis and not the MTAG results?
- The traits used in the MR analyses seem very cherry-picked. Can the authors provide justification why they used hair colour and longevity in their MR analyses? And why they did not investigate traits like cardiovascular disease and type 2 diabetes that they mention in the first paragraph of the introduction and which show strong genetic correlations?
- When investigating the relationship between hair colour and puberty timing, the authors state that "this sex difference appears following the progressive darkening of hair and skin colour during adolescence". Therefore, could the timing of puberty cause darkening of the hair and/or skin? Would it be worthwhile conducting a bidirectional MR analysis to investigate whether this is possible?
- Why were the 5 SNPs identified in a 2010 meta-analysis of 23andMe data used in the sensitivity analysis of hair colour and puberty timing? Could the larger number of SNPs that reached genome-wide significance in 23andMe in Hysi et al. (with corresponding effect sizes) be used instead as this would provide a more powerful instrument?
- In supplementary table 14, it looks like significant heterogeneity is identified in the MR analyses

with prostate cancer. Some discussion of this should be presented.

- In the discussion, the authors mention that SRD5A2 (MEMO1) is a newly highlighted gene implicated in pubertal timing, the disruption of which leads to male-specific reproductive disorders. However, it is actually more strongly associated with AAM than voice breaking in 23andMe. Could the authors please provide more explanation?

Minor

- A number of the values in column Y of Supplementary Table 2 (Concordant direction (with Menarche) for the UKBB FH (Early) effect) are incorrect. For example, rs6931884 in row 3 is not concordant (increasing age at AM and increasing early facial hair).

- In the results section the authors say all 78 SNPs were used in the ALSPAC replication analysis, but in the methods section they state that only 73 SNPs were used. Which is correct?

- Including the gene names in the supplementary tables will allow for easier comparison with the figures and text.

- Can the authors provide a reference for the statement, "Two variants located near to genes that are disrupted in rare disorders of puberty...?"

- The supplementary tables and figures could to be described in more detail. For example, what are the axes on each figure? What is the colour coding for each figure (for example, SF2)? What are each of the columns in the tables?

Reviewer #2:

Remarks to the Author:

In this paper, the authors perform an unprecedentedly high-powered GWAS of puberty timing in males by combining summary statistics for several related phenotypes using MTAG (Turley et al. 2018). This analysis produces results very similar to those found in a previous study of puberty timing in females (Day et al. 2017), with a single exception where an association has a significant effect in the opposite direction. They find a substantial amount of genetic correlation between puberty timing and several other complex traits. They follow up on some of these in Mendelian Randomization (MR) analyses.

I overall thought the project was well-executed and well-written. The comparison of their results to the previous work in females I found especially interesting. In contrast, I thought the MR analyses were quite weak and should either be greatly revised or dropped from the paper. Beyond this, I only have a few other concerns. I go into further detail on these concerns below.

Major Points:

1) I found the MR work in the paper to be very unconvincing. To be ready for publication, the text would require substantial softening of the language around the credibility of the results, a much more thorough explanation of the assumptions of the approach and why readers should expect them to be satisfied, or the analyses should be removed entirely.

1a) MR requires strong assumptions about the absence of alternate pathways from the instrumental variable (in this case, the SNPs) to the outcome variable (e.g., timing of puberty) except through the exposure variable (e.g., hair color). MR-Egger and heterogeneity tests rule out certain forms of pleiotropy that would violate this assumption, but not all of them. For example, if a third latent factor causes both the exposure and outcome variables, we would pass both tests, but a causal interpretation would not be appropriate. In the conclusion, the authors even identify potential pathways that may violate this assumptions (androgens vs estrogens). Why should we think these don't exist? These concerns also exist for the MR analyses of the effect of puberty timing on lifespan and prostate cancer. If the SNPs identified in the puberty timing analysis operate by changing the concentration of hormones in the body, how can we be sure that these

hormones are not affecting the outcome traits directly as opposed to through puberty timing?

1b) With respect to the hair color phenotype specifically (which gets much more real estate in the paper), I'm further concerned by the timing. In many individuals, I believe hair color is not determined until the early 20s, which is why the standard survey questions asks for natural hair color at age 20 (Han et al. 2008), whereas puberty has long passed for most by then. While hair color later in life may be highly correlated with early life hair color, it's variability still makes it more difficult to believe that it could be the cause of puberty timing and not potentially the opposite.

1c) It seems that in several cases, the authors use two-sample IV even when there is sample overlap between the two samples. Why do they lead with these results when the standard errors are unreliable? I'm not totally sure, but it's possible that the standard errors are actually too large when there is sample overlap. This would make their standard errors conservative, but if they would like to lead with those results, they should state and justify this claim explicitly.

1d) For the two exemplar traits, how were these selected? From the point of view of the reader, it makes a difference if these were the only two phenotypes considered, or if the authors tried several and reported the ones that seemed most interesting. This is especially the case since the two examples they present are both only marginally significant and wouldn't survive a multiple testing correction.

2) The sign concordance between the results for males and the previous results for females seems very high to me. Is it perhaps due to relatedness between the male and female UKB samples? If you assume the effect size estimates are exactly the same in men and women, given the effect size estimates (with perhaps a winner's curse correction), what is the expected sign concordance?

3) The authors use a clumping algorithm that is just based on distance. This is a bit unusual. I looked up the most recent GWAS papers in Nature Genetics, and they both had some sort of r^2 threshold in addition to a distance threshold (Maguire et al. 2018, Evangelou et al. 2018). I can understand why they would want to make these results comparable to their previous study on puberty timing in females, but it would also be nice to have some sense of if the SNPs would survive an r^2 -based locus definition as well. For example, what's the maximum r^2 between any pair of "independent" SNPs in your analysis?

Minor points:

1) MTAG produces output for each phenotype. It was not clear to me which phenotype was used when the authors referred to the MTAG output. I'm guessing from some of the figure labels that they used the 23andMe phenotype? If so, this should be clarified. It would also be to good justify using this measure over the other phenotypes.

2) On page 6, the authors state, "(LD, conservatively defined as $r^2 < 0.05$)" Should this be " $r^2 > 0.05$ "?

3) On page 13, the authors state, "The effect estimate for the association of each SNP on the trait of interest is then derived using a marginal likelihood function,..." MTAG is a moments-based method and not a likelihood-based method.

4) On page 14, the authors state, "We used a conservative genome-significant P-value threshold of $P < 5 \times 10^{-8}$ to determine significant SNP associations." This is the standard threshold, not a conservative one. I'd drop the word, "conservative".

5) How do you treat red hair in the hair color GWAS? I believe that red hair is produced by a

different biological process than other colors (Han et al 2008).

References:

Turley, P., Walters, R. K., Maghzian, O., Okbay, A., Lee, J. J., Fontana, M. A., ... & Magnusson, P. (2018). Multi-trait analysis of genome-wide association summary statistics using MTAG. *Nature genetics*, 50(2), 229.

Day, F. R., Thompson, D. J., Helgason, H., Chasman, D. I., Finucane, H., Sulem, P., ... & Altmaier, E. (2017). Genomic analyses identify hundreds of variants associated with age at menarche and support a role for puberty timing in cancer risk. *Nature genetics*, 49(6), 834.

Maguire, L. H., Handelman, S. K., Du, X., Chen, Y., Pers, T. H., & Speliotes, E. K. (2018). Genome-wide association analyses identify 39 new susceptibility loci for diverticular disease. *Nature genetics*, 50(10), 1359.

Evangelou, E., Warren, H. R., Mosen-Ansorena, D., Mifsud, B., Pazoki, R., Gao, H., ... & Ng, F. L. (2018). Genetic analysis of over 1 million people identifies 535 new loci associated with blood pressure traits. *Nature genetics*, 50(10), 1412.

Day, F. R., Thompson, D. J., Helgason, H., Chasman, D. I., Finucane, H., Sulem, P., ... & Altmaier, E. (2017). Genomic analyses identify hundreds of variants associated with age at menarche and support a role for puberty timing in cancer risk. *Nature genetics*, 49(6), 834.

Han, J., Kraft, P., Nan, H., Guo, Q., Chen, C., Qureshi, A., ... & Martin, N. G. (2008). A genome-wide association study identifies novel alleles associated with hair color and skin pigmentation. *PLoS genetics*, 4(5), e1000074.

Reviewer #3:

Remarks to the Author:

Hollis, Day and coworkers report the results of a multi-trait genome-wide association study for male puberty timing including more than 200,000 subjects. The study identifies 78 independent loci, 29 of which have not previously been associated with pubertal timing. The genetic correlation with health outcomes is further characterized, with particular emphasis on a potential causal relationship between early onset puberty, prostate cancer risk, and a shorter lifespan.

The paper addresses an important and interesting topic. Previously female pubertal timing has been extensively analyzed, but less is known about the genetic underpinnings of male pubertal timing. Moreover, the current analyses are based on an impressive number of samples.

Nonetheless the phenotype measurements used in the study impose a bit of a challenge. While pubertal timing is a continuous trait, the phenotype assessment is dichotomized and based on recall, i.e. the timing of voice breaking and the appearance of facial hair were assessed based on a scale "younger than average", "about average", and "older than average". The authors have then used a somewhat unusual approach to construct the GWAS-models, i.e. two sets of binary models per phenotype have been analyzed ("younger" vs. "average" and "average" vs "older"), and all four models have then been combined together with a continuous model for voice breaking using Multi-Trait Analysis of GWAS (MTAG). With this setup, the identical control group has been used twice for the voice breaking and the appearance of facial hair analyses.

- What might the impact be of utilizing the identical control groups many times in the MTAG-

setting?

- Has MTAG been validated for a similar analysis strategy before?

- Does the running GWAS analysis of the extremes, i.e. "younger" vs. "older" for both phenotypes and combining the summary stats with the continuous trait analysis yield similar results?

Obviously, such strategy would result in a considerable reduction of the number of samples, but would there be similar signals still?

- Even if the authors have attempted to seek collective confirmation of the 78 signals identified by MTAG by computing a polygenic risk score and testing for association with timing of voice breaking in an independent cohort, one might still attempt to seek confirmation of single loci, i.e. loci that show very modest association in the individuals GWAS-analyses, e.g. rs1979835, rs17193410, rs7136086, rs10110581, rs10164550, rs10765711, by ordinal regression analysis.

- One of the main results is the reported association between pubertal timing and three loci previously linked with pigmentation. To control for population stratification the phenotypic relationship has been assessed in white males of European ancestry only, also including 40 genetic principal components. Because the identification of an association between a phenotype and pigmentation loci raises the question of underlying population structure, the authors could provide additional information on potential genomic inflation, i.e. by providing information on the genomic inflation factor and the qq-plots for the individual GWAS-analyses.

Reviewer 1

- The questions asked of the UK Biobank participants seem very broad. Were they given any indication of what ‘average’ was? Did you use any of the data from the follow-ups to provide some sort of validation of their answer? If so, how did you account for the individuals that changed answers?

Response: *No indication was given on the ‘average’. Reassuringly, the proportion of individuals changing responses between follow-ups was very low (0.5% for facial hair and 0.3% for voice breaking) and concordance between voice breaking and facial hair reports was high (see Table S1). We used the initial questionnaire assessment for all analyses.*

- The SNP effect estimates from the analysis of early voice breaking and early facial hair in the UK Biobank will be in the opposite direction to the other traits. What impact does this have on the MTAG analysis? Would you get the same result if all of the effect estimates were aligned to later pubertal timing?

Response: *We aligned all traits so that the puberty delaying allele was considered the effect allele. This is now stated in the first paragraph of the results (page 5).*

- I was under the impression that MTAG reported the adjusted estimates for all traits included in the analysis, or performed an inverse-variance weighted meta-analysis that accounts for sample overlap in the GWAS results assuming that the GWAS are all measuring the same phenotype. What was used for this analysis?

Response: *The effect estimates reported here are from MTAG inverse-variance weighted meta-analysis models for continuous age at voice breaking (from the 23andMe sample) as the base trait. As indicated on page 7, for the sex-heterogeneity analysis (age at voice breaking versus AAM in females) we used the effect estimates from the 23andMe sample only as these were directly reported on a continuous scale (in years) which is consistent with the AAM variable.*

- The MTAG method will be somewhat new to many readers and makes some fairly strong assumptions. Could some discussion of these assumptions of MTAG be included, and whether the authors think they are (or aren’t) justified in their analysis?

Response: *We have added to the revised manuscript (page 6) that MTAG relies on three key assumptions: 1) the variance-covariance matrix (denoted Ω) is homogeneous across all SNPs for all traits; 2) sampling variation can be ignored; and 3) LD score regression accurately captures the degree of sample overlap. For assumptions 2) and 3), the original MTAG paper reported simulation tests which indicated that even in cases with extreme sample overlap the assumptions are safely held for almost all realistic applications of MTAG. For assumption 1) the original MTAG paper reported that intuitively this will be violated in scenarios where some SNPs influence only a subset of the traits. While it is plausible that some SNPs may influence either onset of voice breaking or facial hair, the authors go on to state that - “even if this assumption is not satisfied, however, we show analytically that MTAG is a consistent estimator and that its effect estimates always have a lower genome-wide mean squared error than the corresponding single-trait GWAS estimates”. The original MTAG manuscript compared traits with R_G 0.67-0.72, which is similar to the range observed in our paper 0.61-0.81).*

- Were the FDR calculations conducted that are recommended by the MTAG authors?

Response: *As stated in response to the previous comment, violation of the MTAG assumption of homogeneity of effect sizes across traits may yield biased test statistics. To mitigate these effects, the MTAG authors developed the ‘maxFDR’ formula which calculates the upper bound for FDR among all traits in the analysis. In our data, this*

calculation yields a maxFDR 0.00079. The original MTAG paper states: “The maxFDR was 0.0014 for DEP, 0.0080 for NEUR, and 0.0044 for SWB. This calculation suggests that the signals are unlikely to be an artifact of the assumption of homogeneous Ω ”. We can therefore conclude that our results are also unlikely to be the result of violation of the homogeneity assumption, given our low maxFDR. This has been added to the revised version of the manuscript on page 6.

- What is the adj_r2 presented in supplementary table 5 for ALSPAC results? If this is the adjusted r2 from the full model, then I think it is slightly misleading as it is the variance explained by all covariates in the model, adjusting for the number of covariates, not just by the allele score.
Response: *We have clarified in Table S5 the r^2 values for both univariate and adjusted models. All values cited in manuscript have been revised.*
- Would the authors please provide some interpretation of the ALSPAC results, particularly regarding the direction of effect for each phenotype? For example, the beta coefficients for the voice breaking phenotype are negative in the early phases but positive in the latter two and positive for age at voice breaking.
Response: *We have clarified in the text (page 7) that the direction of the ALSPAC results is consistent with the polygenic risk score for later puberty timing. We have also clarified the units of responses in Table S5.*
- A number of large GWAS studies with smaller replication samples are using the slope of a regression analysis between the beta coefficients as evidence for replication (see for example the latest GIANT GWAS, PMID: 30124842). Would a similar analysis be useful in ALSPAC, where the age of voice breaking and facial hair appearance measures are more precisely measured, to complement the existing analysis and provide further evidence of whether the SNPs replicate?
Response: *The small replication sample in ALSPAC ($N=2,394$), compared to our discovery sample (205,354 men for continuous pubertal timing), limits the ability to confirm effects of individual SNPs. We therefore tested a polygenic risk score for later puberty timing, as our downstream analyses, i.e. Mendelian randomisation and genetic prediction, is a score based.*
- Why were the 23andMe voice breaking data used for the genetic heterogeneity analysis and not the MTAG results?
Response: *As described above, for the sex-heterogeneity analysis (age at voice breaking versus AAM in females) we used the effect estimates from the 23andMe sample only as these were directly reported on a continuous scale (in years) which is consistent with the AAM variable.*
- The traits used in the MR analyses seem very cherry-picked. Can the authors provide justification why they used hair colour and longevity in their MR analyses? And why they did not investigate traits like cardiovascular disease and type 2 diabetes that they mention in the first paragraph of the introduction and which show strong genetic correlations?
Response: *We explored hair colour following the annotation of novel loci in our discovery MTAG: at least three loci overlapped genes previously associated with pigmentation. The MR analyses are chosen to confirm previous reports based on AAM variants and lifespan (PMID: 28873088) and prostate cancer (PMID: 28436984).*
- When investigating the relationship between hair colour and puberty timing, the authors state that “this sex difference appears following the progressive darkening of hair and skin colour during adolescence”. Therefore, could the timing of puberty cause darkening of the hair

and/or skin? Would it be worthwhile conducting a bidirectional MR analysis to investigate whether this is possible?

Response: *As stated below in response to a similar comment from Reviewer 2, the hypothesis being tested is not whether there is a causal relationship between puberty timing and hair colour, but whether the biology underlying these traits is shared. The manuscript has been amended to state this explicitly. With regard to the suggestion of a bidirectional MR, while this may be informative the summary data for the hair colour GWAS is unfortunately not available publically.*

- Why were the 5 SNPs identified in a 2010 meta-analysis of 23andMe data used in the sensitivity analysis of hair colour and puberty timing? Could the larger number of SNPs that reached genome-wide significance in 23andMe in Hysi et al. (with corresponding effect sizes) be used instead as this would provide a more powerful instrument?

Response: *We used the older 2010 GWAS for hair colour (in 23andMe) as a sensitivity test that avoids any overlap between discovery sample (for hair colour) and voice breaking sample (in UK Biobank). This rationale is described in the text.*

- In supplementary table 14, it looks like significant heterogeneity is identified in the MR analyses with prostate cancer. Some discussion of this should be presented.

Response: *Due to the removal of the two SNPs as discussed above, when we re-ran the prostate cancer analysis, the p-value crept to the other side of significance ($p=0.06$). As a result we have reduced the prominence of the prostate cancer analysis both in the paper and in the abstract. We now feel that commenting on the heterogeneity of a non-significant result may not be of as much interest to a reader.*

- In the discussion, the authors mention that SRD5A2 (MEMO1) is a newly highlighted gene implicated in pubertal timing, the disruption of which leads to male-specific reproductive disorders. However, it is actually more strongly associated with AAM than voice breaking in 23andMe. Could the authors please provide more explanation?

Response: *We did not mean to indicate SRD5A2 as a male-specific signal (the results text on Page 8 describes this also as a AAM signal) and there is no evidence for sex-heterogeneity ($P=0.99$; Table S6). For balance, we have added in the Discussion that higher 5-alpha-reductase activity has been reported in women with polycystic ovary syndrome.*

- A number of the values in column Y of Supplementary Table 2 (Concordant direction (with Menarche) for the UKBB FH (Early) effect) are incorrect. For example, rs6931884 in row 3 is not concordant (increasing age at AM and increasing early facial hair).

Response: *This has been corrected in the revised Supplementary Tables and manuscript (page 6).*

- In the results section the authors say all 78 SNPs were used in the ALSPAC replication analysis, but in the methods section they state that only 73 SNPs were used. Which is correct?

Response: *It is 73. The text has been corrected (page 7).*

- Including the gene names in the supplementary tables will allow for easier comparison with the figures and text.

Response: *Gene names have been added to Supplementary Tables 2, 6 and 7.*

- Can the authors provide a reference for the statement, “Two variants located near to genes that are disrupted in rare disorders of puberty...”?

Response: *The references have been added (page 8).*

- The supplementary tables and figures could to be described in more detail. For example, what are the axes on each figure? What is the colour coding for each figure (for example, SF2)? What are each of the columns in the tables?

Response: *Details have been added to the supplementary tables and figures.*

Reviewer 2

- I found the MR work in the paper to be very unconvincing. To be ready for publication, the text would require substantial softening of the language around the credibility of the results, a much more thorough explanation of the assumptions of the approach and why readers should expect them to be satisfied, or the analyses should be removed entirely.

Response: *We have updated the text describing the MR analyses in response to reviewers' comments to address the specific comments below. Specifically we have revised any text suggesting a direct causal effect of hair colour on puberty timing, while explicitly stating our hypothesis that hair colour and puberty timing share common underlying pathways which MR tests more robustly compared to non-genetic associations (page 9).*

- MR requires strong assumptions about the absence of alternate pathways from the instrumental variable (in this case, the SNPs) to the outcome variable (e.g., timing of puberty) except through the exposure variable (e.g., hair color). MR-Egger and heterogeneity tests rule out certain forms of pleiotropy that would violate this assumption, but not all of them. For example, if a third latent factor causes both the exposure and outcome variables, we would pass both tests, but a causal interpretation would not be appropriate. In the conclusion, the authors even identify potential pathways that may violate this assumptions (androgens vs estrogens). Why should we think these don't exist? These concerns also exist for the MR analyses of the effect of puberty timing on lifespan and prostate cancer. If the SNPs identified in the puberty timing analysis operate by changing the concentration of hormones in the body, how can we be sure that these hormones are not affecting the outcome traits directly as opposed to through puberty timing?

Response: *We used MR to evaluate the association between hair colour-related variants and puberty timing to provide evidence that these traits have shared biological mechanisms. As the reviewer correctly points out, had we been assessing the explicit hypothesis that hair colour causes a change in puberty timing the results would have to be interpreted differently given the possibility of alternate pathways which are unrelated to the instrumental variable. Given the temporal relationship between puberty and hair colour it seems logical that pathways underlying hair colour influence puberty timing, rather than vice versa.*

- With respect to the hair color phenotype specifically (which gets much more real estate in the paper), I'm further concerned by the timing. In many individuals, I believe hair color is not determined until the early 20s, which is why the standard survey questions asks for natural hair color at age 20 (Han et al. 2008), whereas puberty has long passed for most by then. While hair color later in life may be highly correlated with early life hair color, it's variability still makes it more difficult to believe that it could be the cause of puberty timing and not potentially the opposite.

Response: *As described above, the hypothesis being tested here is not of whether hair colour influences puberty, but rather whether there is shared biology. As mentioned in response to a similar comment from Reviewer 1, while a bi-directional MR may still be informative the summary data from the hair colour GWAS is not publically available.*

- It seems that in several cases, the authors use two-sample IV even when there is sample overlap between the two samples. Why do they lead with these results when the standard errors are unreliable? I'm not totally sure, but it's possible that the standard errors are actually too large when there is sample overlap. This would make their standard errors conservative, but if they would like to lead with those results, they should state and justify this claim explicitly.

Response: *For the hair colour MR to which the reviewer refers, the first stage of the investigation was to check whether there was an association using the more robust*

and statistically powered hair colour instrument from Hysi et al. However as mentioned in the manuscript there is sample overlap with UK Biobank being used in the hair colour discovery GWAS (~46% of the total sample). While this does not preclude the use of two-sample MR, we then sought to confirm these findings using the 5-SNP score, which was less powered but contained summary results from non-overlapping samples. Incidentally, this is likely a broader problem in the field when data for 2-sample MRs is based on the results of large scale consortium meta-analyses therefore the lengths that we have gone to in order to address this exceeds usual practice.

- For the two exemplar traits, how were these selected? From the point of view of the reader, it makes a difference if these were the only two phenotypes considered, or if the authors tried several and reported the ones that seemed most interesting. This is especially the case since the two examples they present are both only marginally significant and wouldn't survive a multiple testing correction.

Response: See response to Reviewer 1, above, on the choice of MR traits (prostate cancer and lifespan, to confirm previous analyses that used AAM loci).

- The sign concordance between the results for males and the previous results for females seems very high to me. Is it perhaps due to relatedness between the male and female UKB samples? If you assume the effect size estimates are exactly the same in men and women, given the effect size estimates (with perhaps a winner's curse correction), what is the expected sign concordance?

Response: We do not find the high concordance in directions of effects between sexes to be surprising as studies of rare disorders of puberty and animal models find substantial shared biology.

- The authors use a clumping algorithm that is just based on distance. This is a bit unusual. I looked up the most recent GWAS papers in Nature Genetics, and they both had some sort of r^2 threshold in addition to a distance threshold (Maguire et al. 2018, Evangelou et al. 2018). I can understand why they would want to make these results comparable to their previous study on puberty timing in females, but it would also be nice to have some sense of if the SNPs would survive an r^2 -based locus definition as well. For example, what's the maximum r^2 between any pair of "independent" SNPs in your analysis?

Response: In response to this comment, we have identified correlations between two pairs of voice breaking loci: LD $r^2 = 0.595$ between rs138625771 and rs75602844; LD $r^2 = 0.293$ between rs112881196 and rs72789842. All other SNP pairs have $r^2 < 0.006$. We have revised the paper throughout to clarify that we identify 76 independent signals for male puberty timing (there are still 29 signals not previously associated with puberty in either sex).

- MTAG produces output for each phenotype. It was not clear to me which phenotype was used when the authors referred to the MTAG output. I'm guessing from some of the figure labels that they used the 23andMe phenotype? If so, this should be clarified. It would also be to good justify using this measure over the other phenotypes.

Response: The effect estimates reported here are from MTAG inverse-variance weighted meta-analysis models for continuous age at voice breaking (from the 23andMe sample) as the output trait, as this corresponds directly with the AAM trait studied in females.

- On page 6, the authors state, "(LD, conservatively defined as $r^2 < 0.05$)" Should this be " $r^2 > 0.05$ "?

Response: This has been amended (page 7).

- On page 13, the authors state, "The effect estimate for the association of each SNP on the trait of interest is then derived using a marginal likelihood function,..." MTAG is a moments-based method and not a likelihood-based method.

Response: *The biorxiv version of the MTAG paper stated (page 4): (<https://www.biorxiv.org/content/biorxiv/early/2017/03/20/118810.full.pdf>) To derive the MTAG estimator of the effect of SNP j on trait t , we derive the marginal likelihood function". However, as this line does not appear in their peer reviewed version, we have now removed this phrase from our paper (page 14).*

- On page 14, the authors state, "We used a conservative genome-significant P-value threshold of $P < 5 \times 10^{-8}$ to determine significant SNP associations." This is the standard threshold, not a conservative one. I'd drop the word, "conservative".

Response: *We have removed this description (page 15).*

- How do you treat red hair in the hair color GWAS? I believe that red hair is produced by a different biological process than other colors (Han et al 2008).

Response: *We considered red hair colour as a separate category in our non-genetic analyses in UK Biobank (Table 2). For the genetic analysis, red hair was an ordered categorical outcome, as in the reported GWAS (Hysi et al. 2018) – indeed in that study the identified variants explained 34.6% of the heritability of red hair. This suggests that the genetic score is informative for all hair colours.*

Reviewer 3

- What might the impact be of utilizing the identical control groups many times in the MTAG-setting?
Response: *MTAG uses bivariate LD score regression to account for the effects of sample overlap. As described above in response to similar a question from Reviewer 1, the developers of MTAG ran simulations involving extreme cases of sample overlap, and demonstrated that even in such cases the test statistics are non biased.*
- Has MTAG been validated for a similar analysis strategy before?
Response: *Yes, the original MTAG paper assessed neuroticism and depression in UK Biobank.*
- Does the running GWAS analysis of the extremes, i.e. “younger” vs. “older” for both phenotypes and combining the summary stats with the continuous trait analysis yield similar results? Obviously, such strategy would result in a considerable reduction of the number of samples, but would there be similar signals still?
Response: *The extremes of voice breaking are a very much smaller sample, comprising about 10% of the sample in the case of voice breaking and 20% for facial hair. As the reviewer notes, that analysis would considerably reduce sample size and statistical power.*
- Even if the authors have attempted to seek collective confirmation of the 78 signals identified by MTAG by computing a polygenic risk score and testing for association with timing of voice breaking in an independent cohort, one might still attempt to seek confirmation of single loci, i.e. loci that show very modest association in the individuals GWAS-analyses, e.g. rs1979835, rs17193410, rs7136086, rs10110581, rs10164550, rs10765711, by ordinal regression analysis.
Response: *Unfortunately, the small replication sample in ALSPAC (N=2,394), compared to our discovery sample (205,354 men for continuous pubertal timing), limits the ability to confirm effects of individual SNPs.*
- One of the main results is the reported association between pubertal timing and three loci previously linked with pigmentation. To control for population stratification the phenotypic relationship has been assessed in white males of European ancestry only, also including 40 genetic principal components. Because the identification of an association between a phenotype and pigmentation loci raises the question of underlying population structure, the authors could provide additional information on potential genomic inflation, i.e. by providing information on the genomic inflation factor and the qq-plots for the individual GWAS-analyses.
Response: *We now add information on the reassuringly low LD score regression intercepts across our GWAS models: 23andMe voice breaking=0.9537; UK Biobank early voice breaking=1.0097; UK Biobank late voice breaking=1.0060; UK Biobank early facial hair=1.0218; UK Biobank late facial hair=1.0343.*

Reviewers' Comments:

Reviewer #1:

Remarks to the Author:

The revisions seem appropriate but I still have a couple of points that require further clarification.

1. The authors now state in the results section that 'In call cases the puberty variables were coded such that the later puberty corresponded to positive effect estimates'. This contradicts what they have in the methods section, where the 'about average' group was used as the reference group in both the 'older than average' and 'younger than average' analyses. It also seems to differ to the results in the supplementary tables, where the effect sizes from the early VB and early FH analyses are in the opposite direction to the other traits. The genetic correlations stated in the 'A multi-trait GWAS for puberty timing in men' section are all positive, but in supplementary table 3 the genetic correlations with the early traits are negative. Would the authors please be able to clarify these discrepancies?
2. The new text added to explain the replication results in ALSPAC still doesn't explain why the direction of effect changes over time for voice breaking. Can the authors give any insights into why the direction of effect is positive at some ages and negative at others? Also, 'lower likelihood of attaining voice breaking in ALSPAC boys' could be phrased more accurately to indicate that the allele score is associated with a lower likelihood for voice breaking at age 13. Finally, the authors state that P_{min} (at age 13.1 years old) = 2.0×10^{-8} , but this is different to what is presented in supplementary table 5 ($P = 2.0 \times 10^{-7}$ for the unadjusted test and $P = 1.4 \times 10^{-7}$ for the adjusted test).
3. I may have misinterpreted the use of MTAG, but I thought the results you were presenting were the voice breaking data from 23andMe adjusted for the other traits – therefore the units from both the MTAG analysis and the 23andMe analysis should be the same? If this is the case, and given the list of novel loci is from the MTAG results, I would be interested to know whether there was heterogeneity between the effect sizes from the MTAG analysis of male puberty timing and the effect sizes for age at menarche. Also, I may have missed it, but I could not find anywhere in the paper that said that the MTAG results were the continuous age at voice breaking results after adjusting for the UK Biobank puberty timing traits; I think this should be made clearer for the reader.
4. I agree with reviewer 2's comment regarding pleiotropy in the MR analyses. The authors have appropriately changed the wording around the hair colour MR analysis, but not for the longevity analysis. For example, given the large genetic correlations between puberty timing BMI/growth related traits, could it be possible that the longevity MR result is driven by an association with BMI? I think this should be addressed in the paper and the authors should be cautious using phrases like 'the findings support a causal effect of later puberty timing in males on longer lifespan...'.

Reviewer #2:

Remarks to the Author:

I thank the authors for their many changes in response to my comments and the comments of the other referees. I think the draft is much improved. I still have some concerns related to the Mendelian Randomization (MR) analyses found in the paper. Those concerns are described below.

Major points:

- 1) I appreciate the new text that states that the authors are using MR to test for shared biology between hair color and puberty timing. Given that reviewer 1 and I were both confused by the purpose of this analysis, I would recommend that the authors explicitly state that they are not attempting to establish causality between the two phenotypes in addition to stating that they are using it to establish shared biology. This is important since MR was specifically designed to test for and measure causal relationships, and I'm unaware of other papers that use MR just to establish

shared biology as is done in this paper.

2) I'm still concerned by the strength of the causal language in the longevity analysis. The authors use strong causal language throughout the paper, including the abstract, results section, and discussion. How can the authors be sure that the SNPs considered don't operate on longevity through alternate pathways that violate the exclusion restrictions? They apparently find SNPs that appear to be heterogeneous using MR-PRESSO, which suggests that there maybe horizontal pleiotropy in this case. I understand that they attempt to correct for this by removing SNPs, but the abstract of the MR-PRESSO paper highlights that removing such SNPs does not always correct for potential bias. While the median-based method appears significant when considered alone, it would not survive a Bonferroni correction when grouped with the other two prostate traits. Overall, I find these results unconvincing. If the authors would like to include them, they should minimally outline the assumptions under which they are causal and discuss why or why not the assumptions may be plausible. If the authors choose to omit this MR analysis entirely, I still believe that the other analyses are sufficiently important and interesting and that they represent a substantial contribution to the literature.

Minor points:

3) In the response, the authors state that they removed the passage saying "The effect estimate for the association of each SNP on the trait of interest is then derived using a marginal likelihood function." This passage is still found on page 14.

4) On page 15, the authors have change "We used a conservative threshold..." to "We used a genome-significant threshold..." The more correct language would probably be "We used the genome-wide significant threshold..."

5) On page 7, that authors state, "A polygenic risk score for later puberty timing based on 73 of the male puberty timing signals was associated with lower likelihood of attaining voice breaking in ALSPAC boys" Why not all 76 SNPs? Where some missing from ALSPAC. I couldn't find an explanation for why 3 SNPs were omitted from the polygenic risk score.

Reviewer #3:

Remarks to the Author:

The authors have addressed all the points raised. I have no further comments.

We thank all the reviewers for their time spent helping improve our manuscript. Our responses to the additional comments raised are highlighted below in red.

Reviewer #1 (Remarks to the Author):

The revisions seem appropriate but I still have a couple of points that require further clarification.

1. The authors now state in the results section that 'In all cases the puberty variables were coded such that the later puberty corresponded to positive effect estimates'. This contradicts what they have in the methods section, where the 'about average' group was used as the reference group in both the 'older than average' and 'younger than average' analyses. It also seems to differ to the results in the supplementary tables, where the effect sizes from the early VB and early FH analyses are in the opposite direction to the other traits. The genetic correlations stated in the 'A multi-trait GWAS for puberty timing in men' section are all positive, but in supplementary table 3 the genetic correlations with the early traits are negative. Would the authors please be able to clarify these discrepancies?

RESPONSE: Apologies, we have deleted this phrase and have clarified throughout the direction of any association. In all cases where genetic scores are generated, these are aligned to positive effects on age at puberty.

2. The new text added to explain the replication results in ALSPAC still doesn't explain why the direction of effect changes over time for voice breaking. Can the authors give any insights into why the direction of effect is positive at some ages and negative at others? Also, 'lower likelihood of attaining voice breaking in ALSPAC boys' could be phrased more accurately to indicate that the allele score is associated with a lower likelihood for voice breaking at age 13. Finally, the authors state that P_{min} (at age 13.1 years old) = 2.0×10^{-8} , but this is different to what is presented in supplementary table 5 ($P = 2.0 \times 10^{-7}$ for the unadjusted test and $P = 1.4 \times 10^{-7}$ for the adjusted test).

RESPONSE: We have clarified in Table S5 which puberty timing outcomes are based on age-combined data and which are age-specific time-points. The GRS for later VB is consistently positively associated age-combined outcomes (e.g. age at peak height velocity) and is consistently negatively associated with attainment of milestones at each specific age. We have clarified this, and amended the cited values, in the main text (Page 7, para 2).

3. I may have misinterpreted the use of MTAG, but I thought the results you

were presenting were the voice breaking data from 23andMe adjusted for the other traits – therefore the units from both the MTAG analysis and the 23andMe analysis should be the same? If this is the case, and given the list of novel loci is from the MTAG results, I would be interested to know whether there was heterogeneity between the effect sizes from the MTAG analysis of male puberty timing and the effect sizes for age at menarche. Also, I may have missed it, but I could not find anywhere in the paper that said that the MTAG results were the continuous age at voice breaking results after adjusting for the UK Biobank puberty timing traits; I think this should be made clearer for the reader.

RESPONSE: We have clarified that our aim was to enhance power to identify GWAS signals for continuous age at voice breaking (as recorded in 23andMe) by combining data from UK Biobank puberty timing traits using MTAG (Page 6, last line).

The plots below show that, while MTAG substantially enhances statistical power of signals for continuous age at voice breaking in 23andMe, unfortunately the effect sizes from the MTAG analysis are markedly attenuated, likely due to the combination of continuous and dichotomous outcome variables. The figure shows consistent attenuation of Beta's between 23andMe and MTAG, which indicates that the MTAG assumption of "homogeneity of variance-covariance matrix for effect sizes" still holds. We have therefore relied on the effect sizes observed in 23andMe.

4. I agree with reviewer 2's comment regarding pleiotropy in the MR analyses. The authors have appropriately changed the wording around the hair colour MR analysis, but not for the longevity analysis. For example, given the large genetic correlations between puberty timing BMI/growth related traits, could it be possible that the longevity MR result is driven by an association with BMI? I think this should be addressed in the paper and the authors should be cautious using phrases like 'the findings support a causal effect of later puberty timing in males on longer lifespan...'.

RESPONSE: We have amended the phrasing of these results to indicate that puberty timing and lifespan are on the same causal pathway.

Reviewer #2 (Remarks to the Author):

I thank the authors for their many changes in response to my comments and the comments of the other referees. I think the draft is much improved. I still have some concerns related to the Mendelian Randomization (MR) analyses found in the paper. Those concerns are described below.

Major points:

1) I appreciate the new text that states that the authors are using MR to test for shared biology between hair color and puberty timing. Given that reviewer 1 and I were both confused by the purpose of this analysis, I would recommend that the authors explicitly state that they are not attempting to establish causality between the two phenotypes in addition to stating that they are using it to establish shared biology. This is important since MR was specifically designed to test for and measure causal relationships, and I'm unaware of other papers that use MR just to establish shared biology as is done in this paper.

RESPONSE: Where genetic instruments represent complex traits, it is common to interpret findings as representing shared biological processes, rather than as 'changing X results in changes in Y'. A notable example of this is: Adult height to coronary artery disease (Nelson et al. NEJM 2015 PubMed ID 25853659), which concludes: "the majority of the relationship is likely to be determined by shared biologic processes that determine achieved height and atherosclerosis development".

2) I'm still concerned by the strength of the causal language in the longevity analysis. The authors use strong causal language throughout the paper, including the abstract, results section, and discussion. How can the authors be sure that the SNPs considered don't operate on longevity through alternate pathways that violate the exclusion restrictions? They apparently find SNPs that appear to be heterogeneous using MR-PRESSO, which suggests that there maybe horizontal pleiotropy in this case. I understand that they attempt to correct for this by removing SNPs, but the abstract of the MR-PRESSO paper highlights that removing such SNPs does not always correct for potential bias. While the median-based method appears significant when considered alone, it would not survive a Bonferroni correction when grouped with the other two prostate traits. Overall, I find these results unconvincing. If the authors would like to include them, they should minimally outline the assumptions under which they are causal and discuss why or why not the assumptions may be

plausible. If the authors choose to omit this MR analysis entirely, I still believe that the other analyses are sufficiently important and interesting and that they represent a substantial contribution to the literature.

RESPONSE: Regarding puberty timing and longevity, it is highly reassuring that consistent significant results are found using 3 different MR approaches (IVW $P=6.7 \times 10^{-4}$; MR-PRESSO $P=8.3 \times 10^{-4}$; and weighted median $P=0.02$, note the weighted median model is recognised to have lower precision - Ref. 30). We have clarified in the text that only 1 SNP was identified by MR-PRESSO as an outlier (there was another outlier SNP prior to removal of correlated SNPs suggested in the previous revision).

We have added to the Discussion a comment on that horizontal pleiotropy might still not be fully accounted for (although this scenario is unlikely, i.e. if horizontal pleiotropy affected $>50\%$ of variants for puberty timing).

We have also amended the phrasing of these results to indicate that puberty timing and lifespan are on the same causal pathway.

Minor points:

3) In the response, the authors state that they removed the passage saying "The effect estimate for the association of each SNP on the trait of interest is then derived using a marginal likelihood function." This passage is still found on page 14.

RESPONSE: Apologies, this has now been changed to "derived using a moments based function"

4) On page 15, the authors have change "We used a conservative threshold..." to "We used a genome-significant threshold..." The more correct language would probably be "We used the genome-wide significant threshold..."

RESPONSE: This has been corrected

5) On page 7, that authors state, "A polygenic risk score for later puberty timing based on 73 of the male puberty timing signals was associated with lower likelihood of attaining voice breaking in ALSPAC boys" Why not all 76 SNPs? Where some missing from ALSPAC. I couldn't find an explanation for why 3 SNPs were omitted from the polygenic risk score.

RESPONSE: Unfortunately genotypes at the remaining 3 SNPs were unavailable in ALSPAC (Page 16).

Reviewer #3 (Remarks to the Author):

The authors have addressed all the points raised. I have no further comments.

Reviewers' Comments:

Reviewer #1:

Remarks to the Author:

I thank the authors for their changes. I still have concerns regarding the MTAG analysis and the inclusion of the early voice breaking/facial hair GWAS results from UK Biobank, which are in the opposite direction to the continuous GWAS and the late voice breaking/facial hair GWAS. For example, if the A allele at a SNP is associated with later pubertal development then in the continuous and later voice breaking/facial hair GWAS it will have a positive effect size, but in the early voice breaking/facial hair GWAS it will have a negative effect size purely due to the coding of the dichotomous outcome. This could be partly driving the lower beta coefficient for the MTAG analysis in comparison to the 23andMe analysis that is presented in response 3. I have raised this in both of my reviews as I am unsure of the effect that this could have in the MTAG analysis and the authors have simply added and now removed one sentence from the results section ('In all cases the puberty variables were coded such that the later puberty corresponded to positive effect estimates'). I will leave this to the editor to decide whether this is an issue that needs further investigation.

Reviewer #2:

Remarks to the Author:

I believe this is a good paper that contains many important results. My only concerns center around the use of Mendelian Randomization. The authors have responded to my previous comments in their rebuttal and have made several changes to their manuscript. However, I believe that their arguments and these changes do not sufficiently address my concerns. I describe my concerns in more detail below.

1) In response to my Major Point 1, the authors claim that many papers use Mendelian Randomization to investigate shared biology between pairs of traits. They cite Nelson et al. (2015) as an example. I looked up this paper, and I don't believe that the abstract and main text mention Mendelian Randomization once. Did they mean to reference a different paper? I still think it is important to explicitly highlight in the main text that, although they are using Mendelian Randomization, they are not doing so to test or imply a causal link between the pairs of traits.

2) In response to Major Point 2, the authors have added a line to the conclusion stating "horizontal pleiotropy remains a possibility if this affected >50% of variants for puberty timing." They have also added arguments that only one SNP was identified by MR-PRESSO and that the consistency of the different approaches increase the likelihood that the estimates correspond to causal effects. I appreciate this, though I feel like this is still insufficient. Imagine, for example, that SNPs play a causal role in some biological process that independently affects both longevity and puberty timing but that there is no causal link between longevity and puberty. I find such a scenario very plausible. In this scenario, all three methods considered would produce consistent spurious results, similar to what the authors find in this paper. Given this concern, I think my conclusions in Major Point 2 in my previous report still stand. The causal language in the abstract and main text is not currently justified. The authors should either (i) remove the MR analyses about prostate cancer and longevity, (ii) greatly weaken the causal language, or (iii) carefully outline the assumptions of their causal analyses and provide evidence that each of them hold. I find it unlikely that (iii) is possible, but if the authors do (i) or (ii), I think the paper would still represent a strong contribution to the literature.

3) In the response to Minor Point 4, the authors claim to have changed "We used a genome-significant threshold" to "We used the genome-wide significant threshold". They have not.

Reviewer #1 (Remarks to the Author):

I thank the authors for their changes. I still have concerns regarding the MTAG analysis and the inclusion of the early voice breaking/facial hair GWAS results from UK Biobank, which are in the opposite direction to the continuous GWAS and the late voice breaking/facial hair GWAS. For example, if the A allele at a SNP is associated with later pubertal development then in the continuous and later voice breaking/facial hair GWAS it will have a positive effect size, but in the early voice breaking/facial hair GWAS it will have a negative effect size purely due to the coding of the dichotomous outcome. This could be partly driving the lower beta coefficient for the MTAG analysis in comparison to the 23andMe analysis that is presented in response 3. I have raised this in both of my reviews as I am unsure of the effect that this could have in the MTAG analysis and the authors have simply added and now removed one sentence from the results section ('In all cases the puberty variables were coded such that the later puberty corresponded to positive effect estimates'). I will leave this to the editor to decide whether this is an issue that needs further investigation.

RESPONSE: We reassure the reviewer that we have rechecked this point several times. We have reinserted the sentence describing the common alignment of all trait statistics to later timing voice breaking (apologies, it had been inserted in the wrong place and then removed when critiqued as being inaccurate) now specifically in the MTAG Results (Page 6 last line) and Methods (Page 14, para 3) sections.

Assurance that this was done correctly is provided by the universal improvement in statistical power when adding data on those UK Biobank dichotomous traits in MTAG – otherwise, if half of the UK Biobank trait data were directionally wrongly coded, we would expect little overall change and at least some signals to weaken.

Reviewer #2 (Remarks to the Author):

I believe this is a good paper that contains many important results. My only concerns center around the use of Mendelian Randomization. The authors have responded to my previous comments in their rebuttal and have made several changes to their manuscript. However, I believe that their arguments and these changes do not sufficiently address my concerns. I describe my concerns in more detail below.

1) In response to my Major Point 1, the authors claim that many papers use Mendelian Randomization to investigate shared biology between pairs of traits. They cite Nelson et al. (2015) as an example. I looked up this paper, and I don't believe that the abstract and main text mention Mendelian Randomization once. Did they mean to reference a different paper? I still think it is important to explicitly highlight in the main text that, although they are using Mendelian Randomization, they are not doing so to test or imply a causal link between the pairs of traits.

RESPONSE: Nelson et al. NEJM 2015 (PubMed ID 25853659) used a conventional Mendelian Randomization analysis (inverse-variance weighted random effects meta-analysis of β_3 estimates for adult height to coronary artery disease) and for the same reason that the genetic approach is “*unlikely to be confounded by lifestyle or environmental factors*” (Discussion line 4-5). They do discuss the possibility of a direct causal effect of shorter height (“*It is also possible that genetically determined height itself alters lifestyle or behavior, which then affects the risk of CAD*”). However, as is the case in our paper, they considered that the instrumented exposure (adult height) represents such a complex trait that it could instead represent sharing common biological pathways. We quote the following sections:

Supplement 1, Pages 11-12: *To obtain an overall association of genetically determined height with CAD, we then combined β_3 estimates for all 180 analysed SNPs using inverse-variance weighted random effects meta-analysis. In a Mendelian randomisation study, (19) the combined estimate of β_3 would then be considered an estimate of the causal association between height and CAD. However our finding is more complex as β_3 could represent several possibilities including height and CAD sharing common biological pathways or genetically-determined shorter height induces behavioural or lifestyle changes that affect CAD risk. These are discussed further in the main paper (see Main Figure 3).*

Figure 3. Interpreting the Association between Genetically Determined Shorter Height and Increased Risk of CAD.

The main advantage of the genetic approach is that it reduces the likelihood of known and unknown demographic, lifestyle, socioeconomic, or behavioral confounders that have an independent effect on height and the risk of CAD (solid black lines) and could give rise to a false association between the two factors. It is possible that the association between the studied genetic variants and height and the association with CAD are through completely different mechanisms (dashed black lines). However, the more likely scenario on the basis of our findings is that height variants affect biologic pathways, which on the one hand determine achieved height and on the other hand influence the risk of CAD (solid red lines). It is also possible that genetically determined height itself alters lifestyle or behavior, which then affects the risk of CAD (dashed red line).

2) In response to Major Point 2, the authors have added a line to the conclusion stating "horizontal pleiotropy remains a possibility if this affected >50% of variants for puberty timing." They have also added arguments that only one SNP was identified by MR-PRESSO and that the consistency of the different approaches increase the likelihood that the estimates correspond to causal effects. I appreciate this, though I feel like this is still insufficient. Imagine, for example, that SNPs play a causal role in some biological process that independently affects both longevity and puberty timing but that there is no causal link between longevity and puberty. I find such a scenario very plausible. In this scenario, all three methods considered would produce consistent spurious results, similar to what the authors find in this paper. Given this concern, I think my conclusions in Major Point 2 in my previous report still stand. The causal language in the abstract and main text is not currently justified. The authors should either (i) remove the MR analyses about prostate cancer and longevity, (ii) greatly weaken the causal language, or (iii) carefully outline the assumptions of their causal analyses and provide evidence that each of them hold. I find it unlikely that (iii) is possible, but if the authors do (i) or (ii), I think the paper would still represent a strong contribution to the literature.

RESPONSE: The causal scenario that the reviewer describes is, we think, well summarised by Figure 3 from the paper by Nelson (pasted above). We acknowledge that our interpretation of the Mendelian Randomisation analyses should be consistently applied for both findings, i.e. for puberty timing to mortality as for hair colour to puberty timing. We have therefore chosen option (ii) to weaken the causal language consistently throughout the paper.

3) In the response to Minor Point 4, the authors claim to have changed "We used a genome-significant threshold" to "We used the genome-wide significant threshold". They have not.

RESPONSE: Apologies. This is corrected (Page 7, line 3).

Reviewers' Comments:

Reviewer #1:

Remarks to the Author:

I thank the authors for addressing all of my previous concerns; I have no further comments or questions.

Reviewer #2:

Remarks to the Author:

This manuscript is much improved. Thanks to the authors for all the changes they have made to it.

Regarding major point 1 from my previous review, I still think it that the manuscript would resolve much potential confusion if the authors added a single sentence (or a clause) highlighting that, although they are using Mendelian Randomization in the hair color and puberty timing analysis, they are testing a hypothesis of shared biology rather than a causal hypothesis. Since MR is nearly always used to test causal hypotheses, this nuance is likely to be missed by readers if it is not explicitly stated.

That said, I have repeated this recommendation three times already. I will leave it to the editor to decide whether making this change is required.

Other than this single point, I am happy with the authors' response and revisions.

REVIEWERS' COMMENTS:

Reviewer #1 (Remarks to the Author):

I thank the authors for addressing all of my previous concerns; I have no further comments or questions.

Reviewer #2 (Remarks to the Author):

This manuscript is much improved. Thanks to the authors for all the changes they have made to it.

Regarding major point 1 from my previous review, I still think it that the manuscript would resolve much potential confusion if the authors added a single sentence (or a clause) highlighting that, although they are using Mendelian Randomization in the hair color and puberty timing analysis, they are testing a hypothesis of shared biology rather than a causal hypothesis. Since MR is nearly always used to test causal hypotheses, this nuance is likely to be missed by readers if it is not explicitly stated.

That said, I have repeated this recommendation three times already. I will leave it to the editor to decide whether making this change is required.

Other than this single point, I am happy with the authors' response and revisions.

RESPONSE: We thank the reviewer for their persistence to make this issue even clearer in the manuscript. We now introduce this hypothesis as the rationale for Mendelian randomisation where we first describe the use of Mendelian randomisation in the results (Page 9, last Para): “To test the shared biological basis between these two phenotypes...”